# Targeted therapies of inflammatory diseases with intracellularly gelated macrophages in mice and rats

Cheng Gao[1,2,4], Qingfu Wang[1,4], Yuanfu Ding[1,3,4], Cheryl H. T. Kwong[1], Jinwei Liu[1], Beibei Xie[1], Jianwen Wei[1], Simon M. Y. Lee[1,2], Greta S. P. Mok[2,3] & Ruibing Wang [1,2] ✉

Membrane-camouflaged nanomedicines often suffer from reduced efficacy caused by membrane protein disintegration and spatial disorder caused by separation and reassembly of membrane fragments during the coating process. Here we show that intracellularly gelated macrophages (GMs) preserve cell membrane structures, including protein content, integration and fluidity, as well as the membrane lipid order. Consequently, in our testing GMs act as cellular sponges to efficiently neutralize various inflammatory cytokines via receptor-ligand interactions, and serve as immune cell-like carriers to selectively bind inflammatory cells in culture medium, even under a flow condition. In a rat model of collagen-induced arthritis, GMs alleviate the joint injury, and suppress the overall arthritis severity. Upon intravenous injection, GMs efficiently accumulate in the inflammatory lungs of acute pneumonia mice for anti-inflammatory therapy. Conveniently, GMs are amenable to lyophilization and can be stored at ambient temperatures for at least 1 month without loss of integrity and bio-activity. This intracellular gelation technology provides a universal platform for targeted inflammation neutralization treatment.

Cytokines play an important role in the pathogenesis of inflammatory diseases, such as rheumatoid arthritis (RA) and acute pneumonia (AP)[1–3]. Biologics, including antagonists for tumour necrosis factor alpha (TNF-α) and interleukin-1β (IL-1β), have achieved some success in anti-inflammatory treatment[4]. However, pathological inflammation is always orchestrated by the synergistic effects of a large number of inflammatory factors, thus inhibition of one or a few may not be sufficient to reverse the inflammation progression[5], because it is practically impossible to employ the mixture of all single-target antibodies in a treatment. In response to the complexity and heterogeneity of inflammatory network, biomimetic nanomedicine has emerged and developed rapidly in recent years. In particular, immune cell membrane-coated nanoparticles were shown to target inflammation and neutralize inflammatory cytokines[6,7]. Because there are various inflammation factor-related receptors on immune cell membrane, these membrane coated nanoparticles inherit the antigenic profile of the source cells, enabling them to act as decoys that can specifically bind and neutralize multiple pathological molecules[8,9]. For example, macrophages (MAs) membrane coated nanoparticle have demonstrated the ability to efficiently bind plaque inflammatory cytokines, and regressed atherosclerosis[10]. Nevertheless, membrane protein disintegration, loss and spatial disorder are inevitable problems during the construction of membrane camouflaged nanomedicine[11,12], which would reduce the specific adsorption and binding of inflammatory cytokines by immune cell membrane-coated nanoparticles, leading to compromised cytokine neutralization and inflammation-targeting efficiency[13,14].

[1]State Key Laboratory of Quality Research in Chinese Medicine, Institute of Chinese Medical Sciences, University of Macau, Taipa, Macao 999078, China. [2]MoE Frontiers Science Center for Precision Oncology, University of Macau, Taipa, Macao 999078, China. [3]Biomedical Imaging Laboratory (BIG), Department of Electrical and Computer Engineering, University of Macau, Taipa, Macao 999078, China. [4]These authors contributed equally: Cheng Gao, Qingfu Wang, Yuanfu Ding. ✉e-mail: rwang@um.edu.mo

On the other hand, live immune cells, with the integral cell membranes and innate cellular functions, possess inherent advantages as potential drug carriers as they can penetrate physiological barriers and accumulate in inflammatory tissues in response to inflammatory signals[15,16]. Thus, immune cells have recently been investigated as the next generation drug carriers[17,18]. A common strategy to load drugs or nanomedicines is via direct drug internalization by phagocytic immune cells[19]. However, intracellular drugs may be metabolized in the complexed cellular environment to induce cytotoxicity and affect cell function-based drug delivery efficiency[20]. In addition, due to the strong efflux capability of immune cells, premature drug leakage can also lead to a low drug utilization[21]. Alternatively, drugs or nanomedicines can be attached onto immune cell surface via covalent conjugation, ligand-receptor binding, or host-guest interaction[22,23]. Nevertheless, endocytosis could not be avoided during the storage and in vivo delivery, resulting from the strong phagocytic ability of immune cells[24], thereafter the same intracellular metabolization and intoxification would occur. Moreover, after targeted accumulation in lesion tissue, these live immune cell-based carriers could be activated in the inflammatory microenvironment and further release cytokines, which may in turn aggravate the inflammation level[25].

Here, we develop intracellularly gelated macrophages (GMs) with highly preserved cell membrane structures, which can function as a natural antiinflammation agent and stable drug carrier at the same time to effectively treat inflammatory diseases. Inside GMs, a supramolecular hydrogel serves as an cytoplasmic skeleton (Fig. 1A). GMs exhibit a significant anti-inflammatory activity in the mouse model of rheumatoid arthritis (RA) by serving as a cellular sponge to efficiently neutralize multi-target inflammatory cytokines (Fig. 1B). Furthermore, GMs also show targeted accumulation in the inflammatory lungs of acute pneumonia (AP) mice upon *i.v.* administration, leading to excellent therapeutic effects (Fig. 1C). Therefore, GMs possess a great potential for clinical translation for targeted anti-inflammatory therapy.

## Results

### Construction of GMs

As the binding ratio between cucurbit[8]uril (CB[8]) and phenylalanine (Phe) is 1: 2, CB[8] may serve as a stable crosslinker to induce the supramolecular gelation of phenylalanine-grafted chitosan (Phe-CS). As shown in Supplementary Fig. 1A, Phe-CS was synthesized via an amidation reaction between Phe and CS. [1]H NMR spectroscopy showed the characteristic resonances of Phe at 7.23 and 7.17 (Supplementary Fig. 1B), indicating that Phe was successfully grafted onto the sidechain of CS, and the grafting yield was ~15.3%. When the concentration of Phe-CS was >1% (wt%) in the presence of CB[8], a good hydrogel morphology was obtained, and the storage modulus increased along with the increasing concentration of Phe-CS (Fig. 2A). Subsequently, 2% (wt%) of Phe-CS was selected as the optimized concentration for further hydrogel characterizations, and the frequency sweep experiments showed a good crosslinking behaviour after addition of 50 μM of CB[8] (Fig. 2B). SEM imaging confirmed the three-dimensional

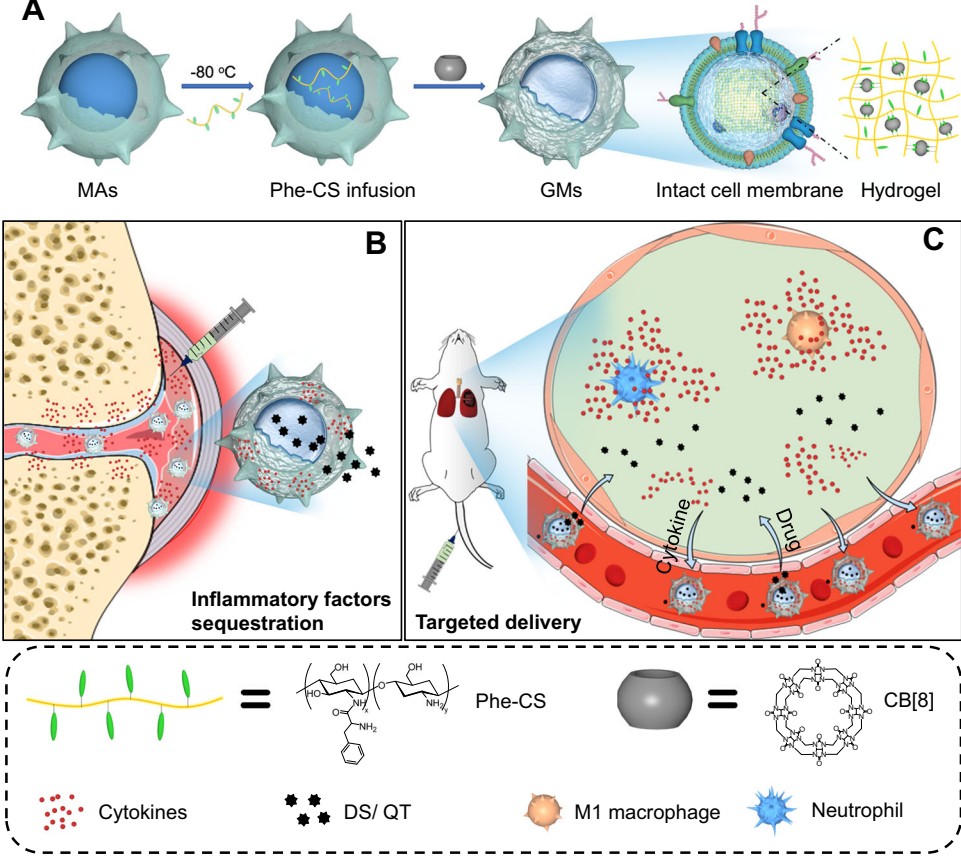

**Fig. 1 | Schematic illustration of GMs construction for targeted anti-inflammatory treatment. A** Preparation of GMs. MAs are infused with phenylalanine-grafted chitosan (Phe-CS) after a freeze-thaw cycle, cucurbit[8]uril (CB[8]) is subsequently added as a crosslinker to construct GMs via intracellular host-guest interactions between CB[8] and Phe in a ratio of 1:2 induced crosslinking and gelation. GMs retain the intact cell membrane structure, including protein content, integrity and fluidity, as well as the membrane lipid order, identical with those of the source MAs. **B** GMs extend the retention time in the inflammatory paw of RA rats upon local administration, and act as a cellular sponge to inflammatory cytokines, to alleviate the joint injuries in both phylactic and therapeutic treatments, which may be further strengthened by intracellularly loaded diclofenac sodium (DS). **C** GMs show targeted accumulation in the inflammatory lungs of AP mice, upon *i.v.* injection, and jointly alleviate the lung inflammation together with intracellular loaded quercetin (QT).

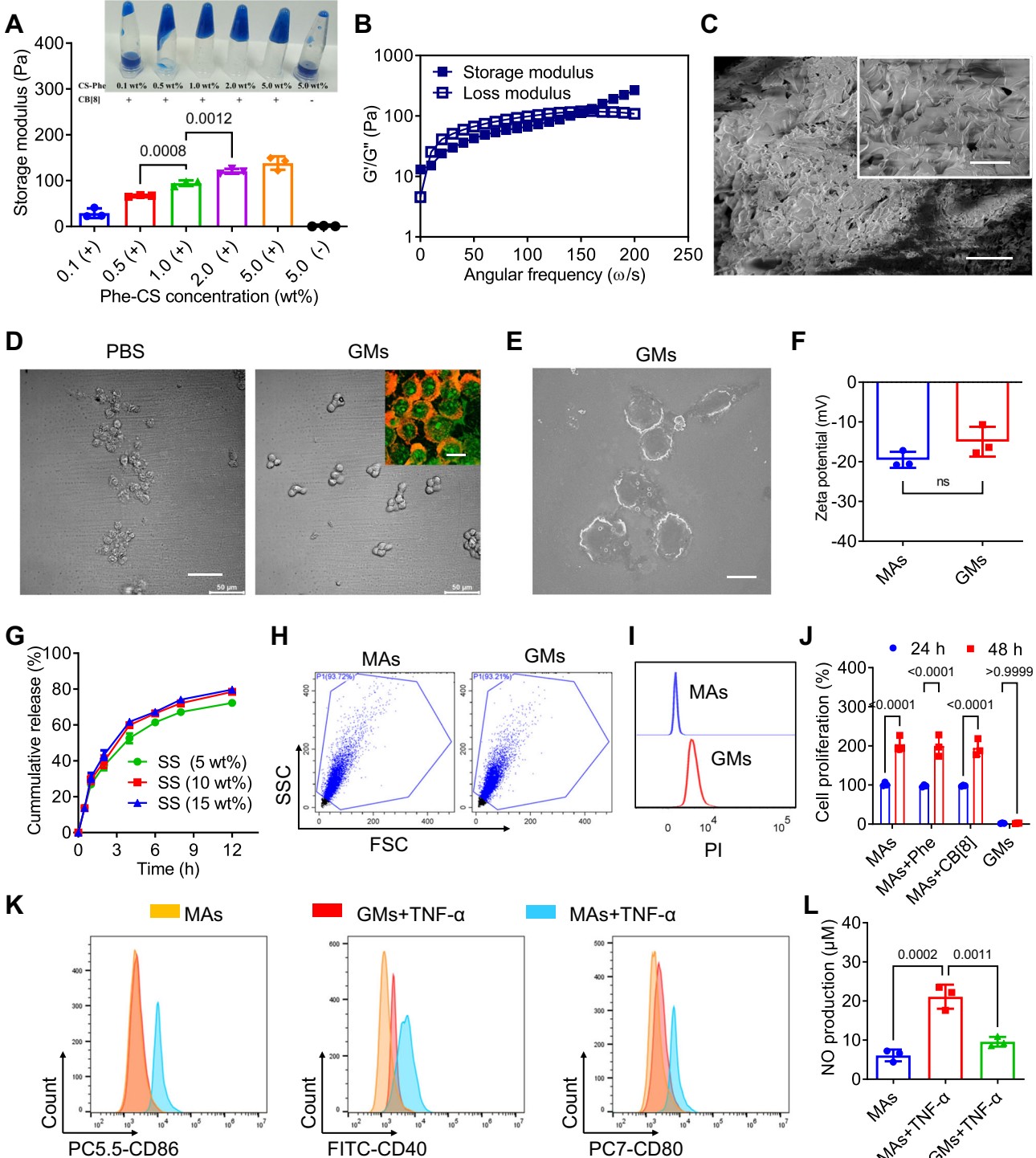

**Fig. 2 | Preparation and in vitro assessment of GMs. A** Storage modulus of hydrogels prepared from different concentrations of Phe-CS (wt%, 0.1%, 0.5%, 1%, 2% and 5%) in the presence of 50 μM of CB[8]. 5% of Phe-CS without addition of CB[8] served as the control group. **B** Frequency sweep experiments on the rheology of hydrogel prepared from 2% (wt%) of Phe-CS and 50 μM of CB[8]. **C** Morphology of hydrogel under SEM. Scale bar: 200 μm. Insert: amplified SEM image. Scale bar: 40 μm. **D** Morphology of GMs under microscopy. MAs in PBS treated with a freeze-thaw cycle served as the control group. Scale bar: 50 μm. Insert: Confocal fluorescence image of membrane dye (DiI) stained GMs, with intracellular hydrogel obtained from CB[8] and FITC conjugated Phe-CS. Scale bar: 10 μm. **E** SEM image of GMs. Scale bar: 10 μm. **F** Zeta potential of MAs and GMs. **G** Drug release behaviours of DS loaded GMs at different drug loading efficiencies (5%, 10%, and 15%). **H** Scatter plots of MAs and GMs determined by flow cytometry. **I** Fluorescence image of PI stained GMs. **J** Cell viability of MAs without any treatment, MAs treated with 1% (wt %) of Phe and 50 μM of CB[8], respectively, and GMs, measured at 24 h and 48 h. **K** The ratio of proinflammatory polarization (CD86+, CD40+ and CD80+ cells) was detected in MAs and GMs after treatment with TNF-α for 12 h by flow cytometry. **L** NO production was analyzed by NO assay kit. All data was presented as mean ± s.d. ($n = 3$). Representative photos in (**C**–**E**) came from three independent experiments on three different samples ($n = 3$). All statistical analyses were conducted using One-Way ANOVA.

network structure of optimized hydrogel (Fig. 2C). As shown in Supplementary Fig. 1C, when the concentration of the crosslinker (CB[8]) was overdosed (e.g. 50 μM), the hydrogel swelling degree decreased with the increasing concentration of Phe-CS. This is because when the concentration of chitosan increases, the hydrogen bond associated structure would easily form between side chains of chitosan, which may function as crosslinking point in the crosslinking network structure. Freshly prepared hydrogel from 2% (wt%) of FITC conjugated Phe-CS (Supplementary Fig. 2A, B) and 50 μM of CB[8] was further utilized to determine the swelling behaviour by fluorescence correlation spectroscopy (FCS) analysis. In comparison to the emission curve of FITC conjugated Phe-CS solution at an excitation wavelength of 480 nm, hydrogel exhibited a red shift from 492 nm to 574 nm (Supplementary Fig. 2C), confirming successful hydrogel formation and labelling. After storage in serum for different durations (0, 1, 3, 5 and 7 days), a very minor and nearly negligible blue shift was observed in the hydrogel's emission curve, indicating high stability of the freshly prepared hydrogel. Furthermore, plot scattering of GMs, analyzed by flow cytometry (Supplementary Fig. 2D), exhibited similar FSC-SSC distribution to that of MAs even after storage in serum for 7 days, suggesting that the size and granularity of GMs did not change. These results further confirmed the morphological stability of GMs in serum.

To enable intracellular gelation inside MAs, adequate cytoplasmic permeation of Phe-CS is critical, therefore a relatively high concentration of Phe-CS (2%, wt%) was introduced in MAs through membrane penetration after a freeze-thaw cycle. Extracellular Phe-CS was removed via centrifugation and washing, subsequently CB[8] was induced into MAs to trigger intracellular gelation, yielding precipitated GMs. GMs under microscopy showed an intact cell morphology, whereas the source MAs without intracellular gelation in PBS showed obvious wrinkles and collapsed cellular skeletons after a freeze-thaw cycle (Fig. 2D). Furthermore, DiI stained MAs were employed for intracellular gelation to allow visualization of the cell membranes. Following conjugation of FITC onto Phe-CS (Supplementary Fig. 2A, B) and subsequent intracellular gelation, GMs showed distinctive green fluorescence (of FITC) in the cytoplasmic hydrogel and red fluorescence (of DiI) in the membrane, displaying a cellular structure of substrate-supported lipid membranes (Fig. 2D). Scanning electron microscopy (SEM) also exhibited highly preserved cell morphology of GMs (Fig. 2E). The zeta potential of GMs was nearly identical to that of MAs, indicating that intracellular gelation process did not affect the cell-surface (Fig. 2F). Collectively, GMs were successfully constructed via the sequential intracellular uptake of Phe-CS and CB[8] after a freeze-thaw cycle, to trigger intracellular supramolecular gelation, the facile process of which certainly allows for large-scale production. Furthermore, drug molecules can be loaded into GMs if a free drug is added together with Phe-CS during the intracellular gelation process. For instance, diclofenac sodium (DS), a nonsteroidal anti-inflammatory drug, was successfully loaded into GMs to afford DS-GMs. As a small-molecule drug, DS may slowly diffuse out from GMs into the extracellular environment. Accordingly, the accumulated drug release from GMs reached ca. 80% after incubation in PBS for 12 h (Fig. 2G).

As activation of MAs by inflammatory environment would aggravate the inflammation, the cell viability and activation status of GMs were assessed. Scattered dot plots of MAs and GMs via flow cytometry analysis further supported the preservation of cell structures after intracellular gelation (Fig. 2H). Meanwhile, GMs displayed strong fluorescence of PI staining, confirming that GMs were not viable (Fig. 2I). Proliferation result showed that intracellular gelation of MAs resulted in complete cell death (Fig. 2J). When incubated in the inflammatory environment (10 μM of TNF-α) for 12 h, the ratios of CD86$^+$ macrophage, CD40$^+$ macrophage and CD80$^+$ macrophage increased, indicating proinflammatory polarization of macrophage after TNF-α treatment. In contrast, TNF-α treated GMs exhibited nearly no CD86$^+$, CD40$^+$ and CD80$^+$ macrophages, comparable to that of

unpolarized MAs (Fig. 2K and S2E). These results indicated that GMs could not be activated in inflammatory microenvironment, which was further confirmed by the low nitric oxide (NO) production of GMs after TNF-α treatment (Fig. 2L).

## Neutralization of proinflammatory cytokines

To investigate the effect of intracellular gelation on the membrane structure, the membrane protein content of GMs was firstly measured. Coomassie blue staining showed identical membrane protein bands between GMs and MAs (Fig. 3A), and there was no statistically significant difference in the membrane protein content of single cells determined by BCA analysis (Fig. 3B). Furthermore, western blotting confirmed the presence and enrichment of some key inflammatory receptors in the membrane of both GMs and MAs, including tumour necrosis factor receptors (TNFR1 and TNFR2), interleukin 6 receptors (IL-6Rα and IL-6Rβ), and interleukin 1 receptor (IL-1R2) (Fig. 3C, D). Taken together, GMs remained complete surface proteins profiles of the source MAs. In the inflammatory diseases such as RA and AP, TNF-α and IL-1β are known to be engaged in the pathogenesis through regulation of inflammation and autoimmunity[26]. Therefore, the binding capability of GMs to TNF-α and IL-1β were tested. GMs were incubated with medium containing 10 μg mL$^{-1}$ each of TNF-α and IL-1β. The remaining cytokines concentrations of supernatant decreased significantly after incubation with GMs in a dose dependence manner (Fig. 3E). Based on measured binding kinetic profiles, the IC$_{50}$ (half maximal inhibitory concentration) value of GMs was determined to be $5.2 \times 10^5$ cells mL$^{-1}$ for TNF-α and $2.7 \times 10^5$ cells mL$^{-1}$ for IL-1β, respectively, by Hill equation. As LPS treated MAs were polarized into the inflammatory cell type[27], significant releases of TNF-α and IL-1β were observed in MAs after incubation with 100 ng mL$^{-1}$ of LPS for 12 h (Fig. 3F). In contrast, the presence of GMs decreased the levels of these cytokines, with a dose-dependent inhibition effect. Overall, GMs showed significant neutralization efficiency across multiple cytokines due to a variety of cytokine receptors highly preserved in the membrane, and could inhibit inflammatory activation of MAs induced by chemokines such as LPS.

## Intact membrane of GMs enhanced cytokine sequestration

In addition to the cytokine receptors in cell membrane, membrane dynamic structure comprised of lipids and proteins also affect the binding of receptors to cytokines, resulting in intracellular signaling cascades[28]. For example, interleukin-6 (IL-6) binds to a membrane-bound IL-6R and the complex of IL-6 and IL-6R associates with a second receptor subunit called GP130 leading to an onset of intracellular signaling[29] (Fig. 3G). Thus, the membrane fluidity of GMs is also closely related with cytokine binding efficiency, thereby affecting the cytokines neutralization efficiency. In order to evaluate the membrane protein fluidity, GMs was treated with 10 ng mL$^{-1}$ of IL-6 for 12 h, and labelled with green fluorescence gp130 antibody and red fluorescence IL-6R antibody to localize their spatial position. As shown in Fig. 3H, the green fluorescence of gp130 in GMs overlapped well with the red fluorescence of IL-6R, and exhibited a high Pearson's correlation coefficient similar to that of living MAs (Fig. 3I), suggesting that GMs possessed a normal membrane protein fluidity comparable to that of MAs. Meanwhile, paraformaldehyde treated MAs (PFA-MAs), served as a negative control group, showed weak and widespread fluorescence of both gp130 and IL-6, suggesting PFA-MAs had poor membrane fluidity due to the PFA fixation. Moreover, an obvious decrease of IL-6 concentration was observed in the supernatant of GMs treated group (3.3 ng mL$^{-1}$), whereas a much higher concentration of IL-6 remained after PFA-MAs treatment (Fig. 3J). This should be attributed to that the fixed membrane proteins of PFA-MAs weakened the binding efficiency with IL-6. Of note, the remaining IL-6 concentration was more than the initial amount of 10 ng mL$^{-1}$ in the MAs treated medium due to the activation of MAs mediated production of cytokines.

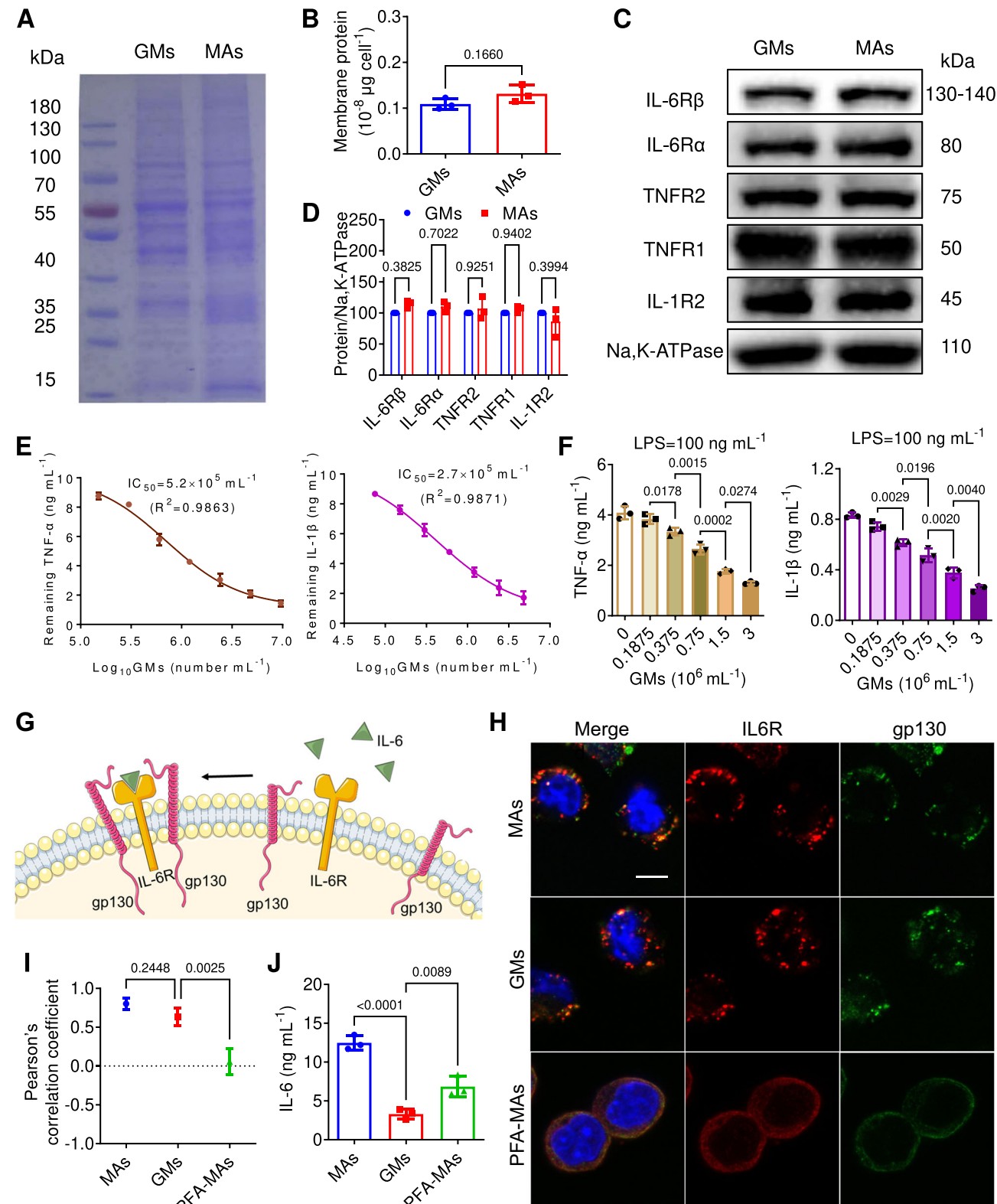

Subsequently, the membrane lipid order, another critical biophysical parameter, influences the dynamics of membrane protein[30]. Laurdan dye staining approach was adopted to study the effect of intracellular gelation on the membrane lipid order[31,32]. After irradiation at 408 nm, the fluorescence intensity of Laurdan stained cell at 450 and 500 nm indicated the degree of ordered membrane and disordered membrane, respectively (Supplementary Fig. 3A). Based on the pixel intensity at cellular periphery, corresponding generalized polarization (GP) values were calculated, and pseudo-coloured GP-intensity-merged images (Supplementary Fig. 3B) and intensity histograms (Supplementary Fig. 3C) were created by ImageJ. The membrane order imaging and GP value of GMs were similar to that of living MAs, confirming the high reservation of plasma membrane order by intracellular gelation. In contrast, living MAs subject to freeze-thaw treatment in the absence of hydrogel components (Non-gelated MAs) showed noticeable alteration in membrane order and GP values,

**Fig. 3 | Membrane protein integrity and fluidity of GMs enhanced cytokine sequestration efficiency. A** Membrane protein bands of MAs and GMs via Coomassie blue staining. **B** Membrane protein contents of MAs and GMs determined by BCA assay. **C** Western blot analysis on the expression of IL-6Rα, IL-6Rβ, TNFR2, TNFR1 and IL-1R2 in the members of MAs and GMs, respectively. **D** Quantitative result of protein expression. **E** Different doses of GMs (0.15, 0.3, 0.6, 1.2, 2.4, 4.8 and 9.6 × 10⁶ mL⁻¹) were incubated with PBS solution containing 10 μg mL⁻¹ each of TNF-α and IL-1β, and the supernatant concentrations of TNF-α and IL-1β were measured by Elisa kits after incubation for 1 h. **F** MAs were co-treated with 100 ng mL⁻¹ of LPS and different dose of GMs (0.1875, 0.375, 0.75, 1.5 and 3, ×10⁶ mL⁻¹) for 12 h, and the supernatant concentrations of TNF-α and IL-1β were measured by Elisa kits. **G** Schematic illustration of IL-6 trans-signalling pathway. **H** MAs, GMs and PFA-MAs

were incubated with 10 ng mL⁻¹ of IL-6 for 12 h, and incubated with rabbit gp130 primary antibody and mouse IL-6R primary antibody, respectively, followed by incubation with anti-rabbit IgG Fab2 Alexa Fluor 488 and anti-mouse IgG Fab2 Alexa Fluor 594, and fluorescence imaging was conducted by confocal microscopy. Scale bar: 5 μm. **I** Pearson's correlation coefficient of fluorescence co-localization was determined by ImageJ. **J** MAs, GMs and PFA-MAs at the number of 3 × 10⁶ mL⁻¹ were incubated with 10 μg mL⁻¹ of IL-6 for 12 h, and the remaining IL-6 concentration was determined by Elisa kit. All data was presented as mean ± s.d. (*n* = 3). Representative photos in (**A**, **C**–**H**) came from three independent experiments on three batches of MAs and GMs (*n* = 3). All statistical analyses were conducted using One-Way ANOVA.

indicating the destruction of membrane lipid order. Furthermore, the neutralization efficiency of GMs towards TNF-α was much higher than that of Non-gelated MAs (Supplementary Fig. 3D), revealing that ordered membrane contributed to the high cytokine sequestration efficiency. Collectively, these results demonstrated that the intracellular gelation process had little influence on the phospholipid bilayers, including membrane protein mobility and lipid order, and thereby ensuring the innate membrane structure of GMs for efficient cytokine neutralization.

Macrophage membrane coated nanoparticles (MM-NPs), gelated red blood cells (GRBCs) and GMs derived from enzymes (pepsin and trypsin) treated macrophages (EGMs) were prepared to test the cytokine neutralization efficiency and compare with that of GMs. GMs, EGMs and MM-NPs prepared from same number of macrophages (3 × 10⁶), and 3 × 10⁶ of GRBCs were incubated in 1 mL of PBS solution containing 10 ng mL⁻¹ of TNF-α and 10 ng mL⁻¹ of IL-1β for 1 h, respectively. As shown in Supplementary Fig. 4A, B, GMs and MM-NPs significantly reduced the TNF-α level into 2.93 and 6.87 μg mL⁻¹, and IL-1β level into 2.36 and 6.14 μg mL⁻¹, respectively, in supernatant. In contrast, GRBCs and EGMs showed negligible effects on the cytokine concentration. Of note, the remaining TNF-α and IL-1β concentrations in the MM-NPs treated group was ~2.45-fold and ~2.6-fold higher than that of the GMs treated group, respectively. These results exhibited a strong cytokine sequestration capability of GMs in comparison with MM-NPs. In addition, the total membrane protein content in GMs and MM-NPs prepared from same number of macrophages (3 × 10⁶) were analyzed via BCA protein assay. As shown in Supplementary Fig. 4C, the total membrane protein content of GMs was 3 × 10⁻³ μg, ~1.86-fold than that of MM-NPs (1.61 × 10⁻³ μg), indicating that the membrane protein loss led to a low cytokine sequestration efficiency. With the 1.86-fold of membrane protein loss, 2.45-fold or 2.6-fold of decreased neutralization efficiency of MM-NP when compared with GMs was likely attributed to the protein spatial disorder during membrane re-assembly process. Collectively, these evidences showed that the intact membrane structure of GMs contributed to a better cytokine neutralization efficiency.

### In vitro inflammatory tropism of GMs

Inflammation involves recruitment of white blood cells to the inflammatory tissue[25]. Due to the intact cell membrane structures, GMs were expected to preserve the inflammatory tropism of MAs. To verify this hypothesis, HUVECs treated with LPS to induce cellular inflammation were subsequently co-cultured with GMs. As shown in Fig. 4A, most GMs with spherical shape (red arrow) were bound with LPS treated HUVECs (dendritic shape) after incubation for 12 h, whereas GMs and normal HUVECs (without LPS treatment) did not couple with each other and only individual cells were observed. To quantify their binding efficiency, HUVECs and GMs were stained with membrane dye DiO (green fluorescence) and DiD (red fluorescence), respectively. As shown in Fig. 4B, DiO stained HUVECs and DiD stained GMs were mainly distributed in the bottom right quadrant and upper left quadrant of scatters via flow cytometry, respectively, in the mixture of GMs

and normal HUVECs. Few data cluster (12.2%) was found in the upper right quadrant of scatters plotting both fluorescence of DiD and DiO, further indicating the separation of both cells (Fig. 4C). In contrast, a new data cluster was observed in the upper right quadrant of scatter by detecting the mixture of GMs and LPS treated HUVECs, and the conjugation efficiency reached 57%, suggestion the strong inflammatory tropism of GMs. Furthermore, in order to investigate the dynamic inflammatory tropism of GMs, a single channel microfluidic chip was developed to simulate blood flow. LPS treated rat synovial cells (RSC-364 cells) were incubated in the groove of the chip (Fig. 4D). FITC labelled GMs were injected into the chip via a syringe pump at a flow rate of 200 μL min⁻¹, and fluorescence imaging showed that green fluorescent GMs were bound with seeded RSC-364 cells after injection for 1 min (Fig. 4E). An increased number of GMs bound to the groove was observed after injection for 2 min (Fig. 4F). Furthermore, a double-channel microfluidic chip was designed for selective binding analysis of GMs. RSC-364 cells with and without LPS pre-treatment were cultured in the bottom and upper channels' grooves, respectively (Fig. 4G). After injection at a flow rate of 200 μL min⁻¹ for 2 min, the green fluorescence of DiO labelled GMs at bottom channel's groove was much higher than that of upper one (Fig. 4H), revealing that much more GMs were bound with LPS treated RSC-364 cells than that to the untreated RSC-364 cells. Similar phenomenon was also observed after injection of DiO stained PFA-MAs (Fig. 4I) or Non-gelated MAs (Fig. 4J), but the absolute green fluorescence intensity in both these two groups in the bottom channels of inflammatory RSC-364 cells was weaker than that of GMs, indicating that GMs had a better inflammatory tropism than PFA-MAs and Non-gelated MAs, attributed to the poor membrane protein fluidity of PFA-MAs and destruction of membrane lipid order of Non-gelated MAs, respectively. The results showed that GMs had a selective binding towards inflammatory cells even under a flow condition, attributed to the intact membrane structures of GMs including protein content, fluidity and lipid order.

### Enhanced local retention and therapy of CIA in rats

RA is a long-term autoimmune disorder characterized by systemic inflammation that primarily affects joints[4]. Considering that TNF-α and IL-1β are the two key pro-inflammatory cytokines in RA[33], the interaction between GMs and inflammatory joint was investigated in a Sprague Dawley (SD) rat model of collagen-induced arthritis (CIA, a commonly used experimental RA model), which was constructed via local injection of complete freund's adjuvant (CFA) for initial immunization and incomplete freund's adjuvant (IFA) for inflammation booster in the footpad of a rear paw[34]. After 4 weeks of development of arthritis, CIA rats were randomly and blindly separated into two groups (*n* = 3 in each group). Red fluorescent Cyanine5 (Cy5) was loaded into GMs (Cy5-GMs) during intracellular gelation in order to track the drug release behaviour and biodistribution in the swelling paw, and the inflammatory footpad of CIA rats were locally administered with free Cy5 and Cy5-GMs at 0.5 mg kg⁻¹ of Cy5, respectively. The fluorescence intensity decreased quickly in the swelling paw of free Cy5 treated rat, whereas Cy5-GMs treated rats maintained strong red fluorescence

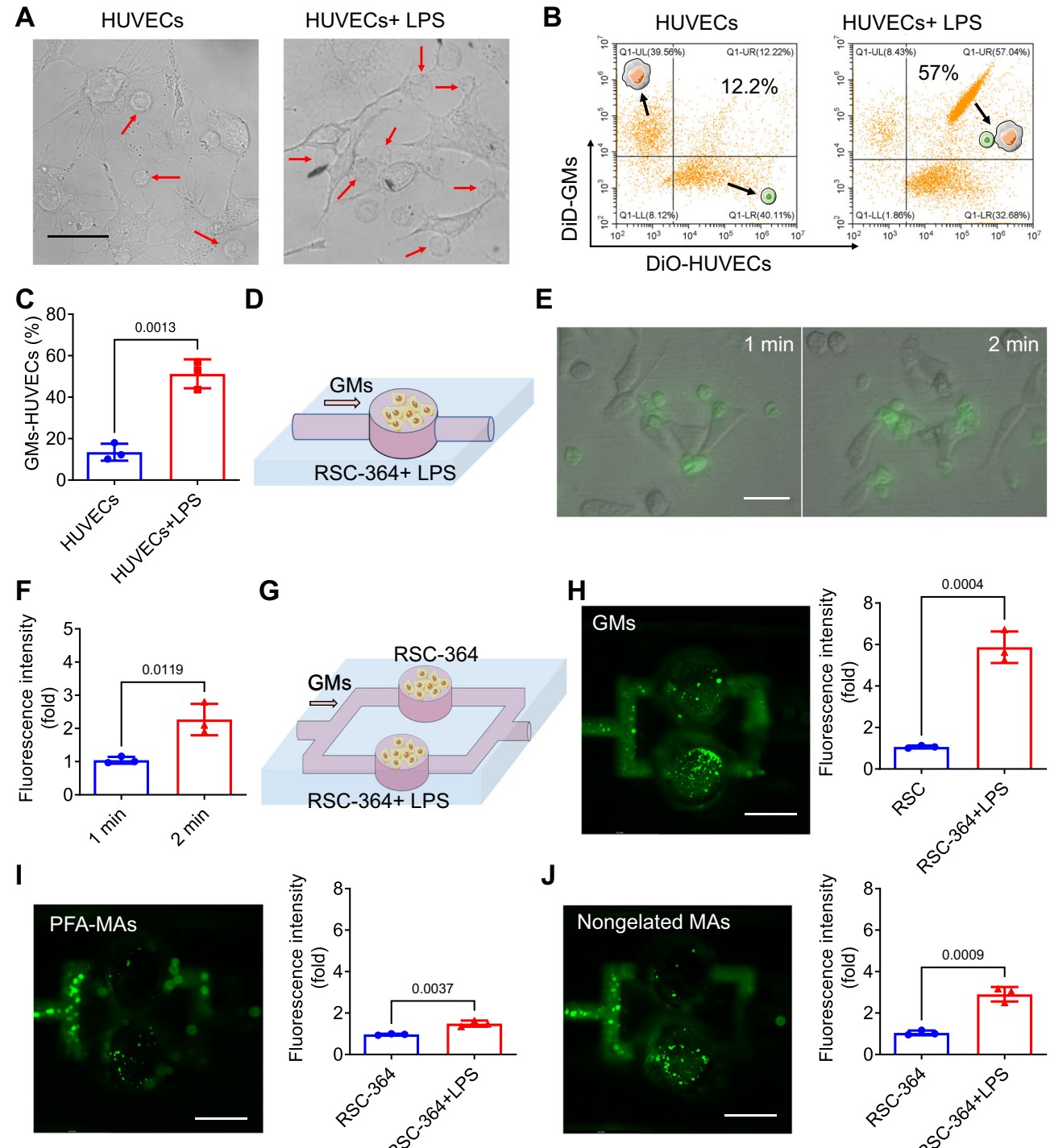

**Fig. 4 | In vitro static and dynamic inflammatory tropism of GMs. A** HUVECs were incubated in culture media with and without 10 µg mL⁻¹ of LPS for 12 h, respectively, and then imaged under microscopy after treatment with GMs for another 12 h. Red arrow refers to as GMs. Scale bar: 30 µm. **B, C** Flow cytometry analysis on the binding efficiency between DiD-GMs and LPS treated DiO-HUVECs or free DiO-HUVECs (**B**) and quantitative result (**C**). **D** Schematic illustration of the dynamic binding of GMs towards inflammatory RSC-364 cells. **E, F** RSC-364 cells were seeded in the groove of microfluidic chip and treated with 10 µg mL⁻¹ of LPS for 12 h, and then FITC labelled GMs were injected into the chip at a flow rate of 200 µL min⁻¹ (*n* = 3). After injection for 1 and 2 min, the single channel chips were fluorescently imaged by confocal fluorescent microscopy (**E**) and quantified by

ImageJ (**F**). Scale bar: 40 µm. **G** Schematic illustration of dynamic binding of GMs towards LPS treated and untreated RSC-364 cells, respectively. **H–J** RSC-364 cells with and without LPS treatment were cultured in the bottom and upper channels, respectively (*n* = 3). After injection of DiO labelled MAs (**H**) PFA-MAs (**I**) and Non-gelated MAs (**J**) for 2 min, the double-channel microfluidic chips were fluorescently imaged by confocal fluorescence microscopy, and quantified by ImageJ. Scale bar: 3 mm. All data was presented as mean ± s.d. (*n* = 3). Representative photos in (**A**) came from independent experiments on three different cell samples (*n* = 3). Statistical analyses of (**C–H**) were conducted by using One-Way ANOVA and Two-Way ANOVA, respectively.

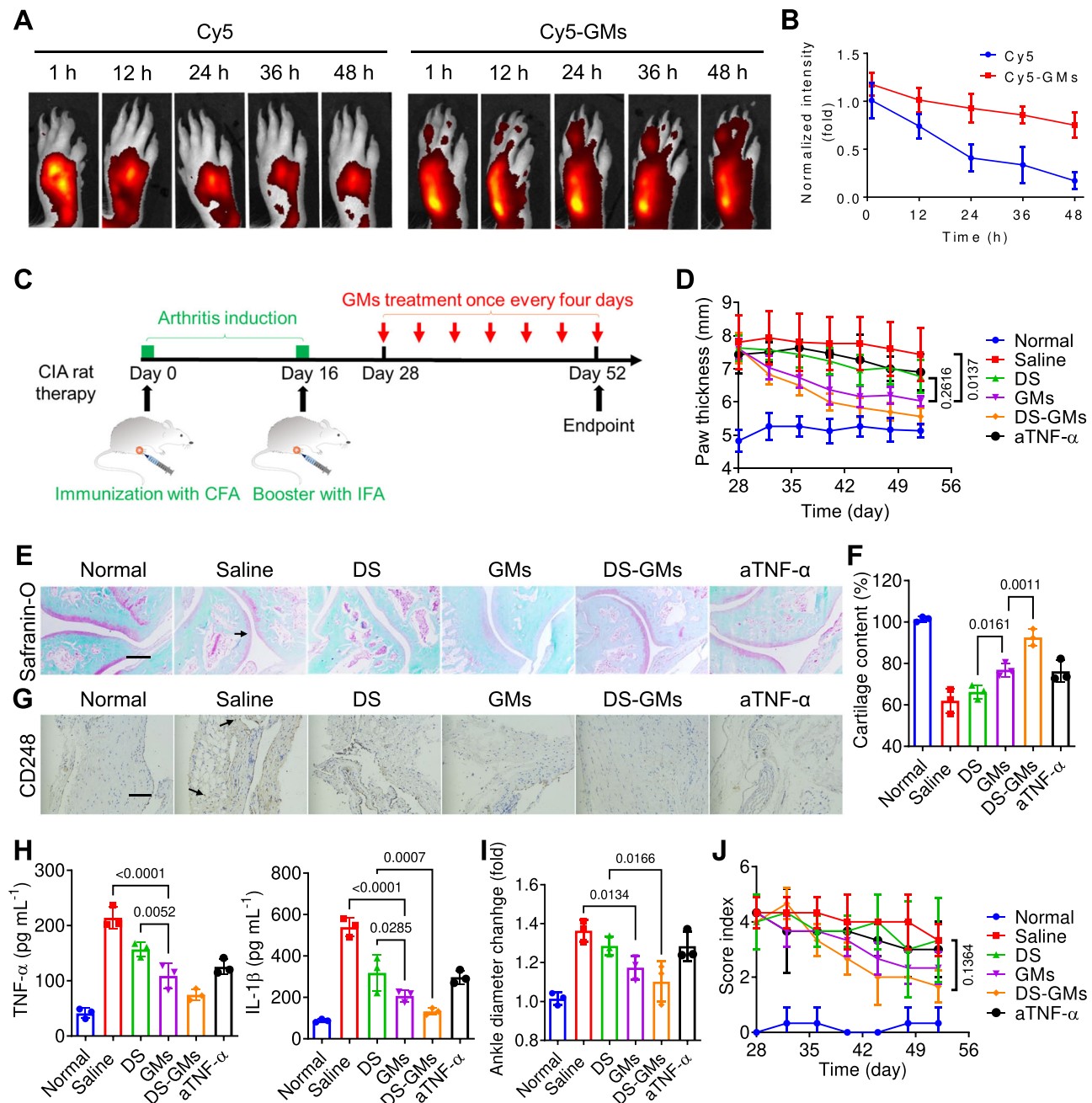

**Fig. 5 | GMs ameliorated the joint injuries in CIA rats via cartilage retention and cytokine neutralization. A** Following arthritis induction, the inflammatory foot-pad of CIA rats was locally administered with free Cy5 and Cy5-GMs at 0.5 mg kg⁻¹ of Cy5 (*n* = 3), and imagined at different time points by in vivo imaging system (IVIS). **B** Quantitative analysis on the fluorescence intensity of the swelling paw. **C** Scheme of the study protocol of a therapeutic regimen with CIA rats. **D** The changes of paw thickness in CIA rats after receiving different treatments. **E, F** Representative images of safranin-O staining on the ankle sections from treated rats (**E**) and quantification for the cartilage content (**F**). Scale bar: 200 μm. **G** CD248 staining on the synovial intimal lining from treated rats. Scale bar: 100 μm. **H** The concentrations of serum TNF-α and IL-1β at the endpoint of treatment. **I** The ankle diameter changes of treated rats compared to that of the normal rats recorded at Day 52. **J** The overall arthritis score of the treated rats recorded every 4 days with a total of 24 days after arthritis induction. Representative photos in (**E**−**G**) came from independent experiments on three different ankles and synovial intimal linings, respectively (*n* = 3). All data was presented as mean ± s.d. (*n* = 3). All statistical analyses were conducted by using One-Way ANOVA.

intensity in the paw for at least 48 h (Fig. 5A). The slow fluorescence attenuation curve of Cy5-GMs treated rats indicated that GMs increased the retention time of intracellularly loaded Cy5 in the swelling paw (Fig. 5B), attributed to the cytokine mediated binding between GMs and the inflammatory domain of swelling paw.

Subsequently, the therapeutic effect of GMs on the CIA joints was further examined, and the footpad of CIA rats were locally administered with saline, DS, GMs, and DS-GMs at 3 × 10⁶ of GMs per rat,

1 mg kg⁻¹ of DS, 2 mg kg⁻¹ of hydrogel, 3 × 10⁶ of MAs once every 4 days until Day 52 (Fig. 5C). As an anti-TNF-α antibody is clinically utilized to block the activity of TNF-α for the treatment of RA's joint swelling and other inflammatory symptoms[35], CIA rats treated with purified anti-rat TNF-α antibody (aTNF-α), and mixture of DS and aTNF-α (DS+aTNF-α) at 1 mg kg⁻¹ of aTNF-α and 1 mg kg⁻¹ of DS served as the control groups. The changes of paw phenotype were imaged during the treatment period (Supplementary Fig. 5), and the paw of saline treated rats

suffered from erythema and severe swelling. In contrast, this effect was significantly lessened for rats treated with GMs, quantified by measuring the paw thickness (Fig. 5D). A modest therapeutic effect was observed in the TNF-α and DS treated groups. As expected, with the combination effects from intracellularly loaded DS, DS-GMs exhibited the best anti-arthritis efficacy. At the endpoint of experiment, Safranin-O staining on the ankle sections showed high expression of sulfated glycosaminoglycans in the cartilage of GMs treated rats than those of aTNF-α and DS groups (Fig. 5E, F). CD248 expression of fibroblast-like synoviocytes (FLSs) in the synovial intimal lining is known to upregulate TNF-α and IL-1β and induce joint damage of RA patients[36]. Immunochemical staining of GMs treated group showed nearly no CD248+ cells in the synovium and only a few CD248+ cells were found in the section of aTNF-α and DS treated groups (Fig. 5G), whereas a large amount of CD248+ cells existed in the synovium of saline treated group. In particular, DS-GMs jointly decreased the level of sulfated glycosaminoglycans in the cartilage and CD248+ cells in the synovium. As GMs are able to neutralize diffusive arthritogenic factors, the systemic inflammatory response was further evaluated. As shown in Fig. 5H, GMs significantly reduced the serum concentrations of TNF-α and IL-1β, which are known to increase with the development of arthritis and correlate strongly with disease severity[37], showing modestly better anti-inflammation effects than aTNF-α. The ankle diameter of rats with local injection of GMs was also comparable to that of rats treated with aTNF-α, but much smaller than that of saline treated rats and higher than that of DS-GMs treated rats (Fig. 5I). Finally, the severity of arthritis can be scored by standard visual inspections, and the arthritis score of CIA was evaluated by assessing the swelling and redness of the rat paws[38]. The results showed that GMs treated rats had a low overall arthritis score in comparison to that of aTNF-α and saline treated groups (Fig. 5J), attributed to that the intact membrane structures, including multiple cytokine receptors, protein fluidity and lipid order, may maximize the neutralization efficiency of GMs towards various cytokines. As shown in Supplementary Fig. 5 and Fig. 6, the therapeutic efficacies of both MAs and hydrogel on CIA rats were similar to that of saline, without obvious improvements in symptoms observed in the joint injury. DS+aTNF-α exhibited better anti-arthritis efficacy than that of DS treatment and aTNF-α treatment, but was still moderately weaker in comparison with DS-GMs, attributed to that GMs increased the retention time of payload in the swelling paw (Fig. 5A, B). MM-NPs prepared from the same number of macrophages ($3 \times 10^6$) alleviated arthritis symptom to a certain extent, but still exhibited a relative worse effect on CIA rats with statistical difference in comparison to that of GMs. This is likely attributed to the membrane protein loss and spatial disorder during MM-NP preparation process, leading to a relative lower cytokine neutralization efficiency. As expected, DS-GMs treated rats exhibited the lowest overall arthritis score, showing the power of intracellular supramolecular gelation technology together with the loaded therapeutic agent for combined anti-arthritis effects. Finally, CIA rats treated with 2 mg kg⁻¹ of prednisolone (PSL, an anti-inflammatory glucocorticoid) served as a positive control group. Moreover, PSL suppressed arthritis significantly, similar to that of DS-GMs. However, glucocorticoid-induced severe adverse effects led to recommendations that glucocorticoids could only be used at low doses over a short period of time[39]. Thus, GMs provides an ideal substitution for safe, long-term treatment of arthritis.

### Phylactic treatment of CIA rats with lyophilized GMs

Most biologics are usually stored in a freezing condition of −80 °C, limiting their transportation and use in remote and under-developed regions. Accordingly, the significant advantage of GMs in storage conditions was identified. We discovered that GMs can be lyophilized into powders as a potential form of product, with ease of storage and transportation under ambient conditions (Fig. 6A). After storage at ambient temperature for 1 month, GM powders were rehydrated in PBS and became well dispersed, whereas Non-gelated MAs through the same lyophilization/rehydration process remained in the form of aggregates after rehydration. The morphology of rehydrated GMs (also referred to as RGMs) maintained the original spherical cell shape (Fig. 6B), and the lyophilization, storage for a month, as well as rehydration process did not affect their cytokine neutralization capability, which was still comparable with that of fresh GMs (Fig. 6C, E). Initiation of treatment in early RA usually improves the success of achieving disease remission and reduces joint damage and potential disability[40,41]. Subsequently, the phylactic treatment of RA by RGMs was further investigated in CIA rats. CIA rats with early-stage arthritis was locally administered with saline, DS, RGMs and rehydrated DS-GMs (also referred to as DS-RGMs) at $3 \times 10^6$ of cells per rat and 1 mg kg⁻¹ of DS (Fig. 6D). Rats treated with 1 mg kg⁻¹ of aTNF-α and normal rats served as the positive and normal control groups, respectively. Similar to the therapeutic treatment of CIA rats described previously, the phenotype of paw swelling was obviously alleviated in the rats treated with RGMs (Supplementary Fig. 7A), and the paw thickness was also reduced (Fig. 6E). DS-RGMs exhibited very modest paw swelling and the lowest paw thickness when compared with other groups. Haemotoxylin and eosin (HE) stained sections from RGMs treated group (Supplementary Fig. 7B) showed an even distribution of chondrocytes in the articulate cartilage without obvious degeneration, and safranin-O staining further confirmed a similar reduction of immune infiltration and preservation of cartilage in the rats treated with RGMs, aTNF-α and DS, respectively (Fig. 6F, G). Fibronectin-positive cells disappeared in the synovium of RGMs, aTNF-α and DS treated groups (Fig. 6H). Furthermore, the systemic inflammation in rats was also efficiently inhibited by RGMs (Fig. 6I), and RGMs treated rats showed a small diameter of ankle below the knee (Fig. 6J). Blinded scoring in the experimental rats showed that GMs treated rats had a relatively lower arthritis score than aTNF-α group and DS treated groups (Fig. 6K). The therapeutic efficacy of RGMs was not diminished after being stored at room temperature for 1 month in a lyophilized form, and the ability to store biologics at room temperature is a significant advance that may facilitate industrial production and clinical translation. Finally, based on results of histological analysis on different organs (Supplementary Fig. 8A), blood cell count (Supplementary Fig. 8B), liver function (Supplementary Fig. 8C) and kidney function (Supplementary Fig. 8D) evaluations, the phylactic treatment with RGMs and DS-RGMs did not induce any pathological damages in CIA rats, showing a safe profile.

### Targeted therapy of AP in mice

Due to the excellent tropism to inflammation, GMs may function as a universal anti-inflammatory agent and delivery platform at the same time. To verify this significant potential, in addition to CIA rat model, GMs were further investigated for targeted treatment of acute lung inflammation, AP, in mice. AP mice were constructed via intratracheal treatment with 50 μL of lipopolysaccharide (LPS). After administration of LPS for 6 h, AP mice were blindly and randomly divided into three groups (n = 3 in each group), and i.v. administered with free Cy5, Cy5-GMs and the physical mixture of Cy5 and GMs (Cy5+GMs) at 0.5 mg kg⁻¹ of Cy5 and $5 \times 10^6$ of GMs, respectively. After injection for 10 min, the collected lung tissues from Cy5-GMs treated mice showed a bright red fluorescence intensity, which was 2.85-fold that of free Cy5 treated mice (Fig. 7A, B). However, Cy5+GMs treated group only showed a fluorescence intensity similar to that of Cy5 treated group. These results indicated that Cy5 loaded GMs increased the accumulation rate of Cy5 in the inflammatory lungs, when compared with free Cy5 and Cy5+GMs, attributed to the inflammatory tropism of GMs. Additionally, GRBCs were supplemented as a non-targeting control gelated cells, and RBCs and macrophages were both stained with membrane red fluorescence dye (DiD) before intracellular gelation. AP mice (n = 3) were i.v. injected with DiD stained GMs (DiD-GMs), DiD stained EGMs (DiD-EGMs) and DiD stained GRBCs (DiD-GRBCs),

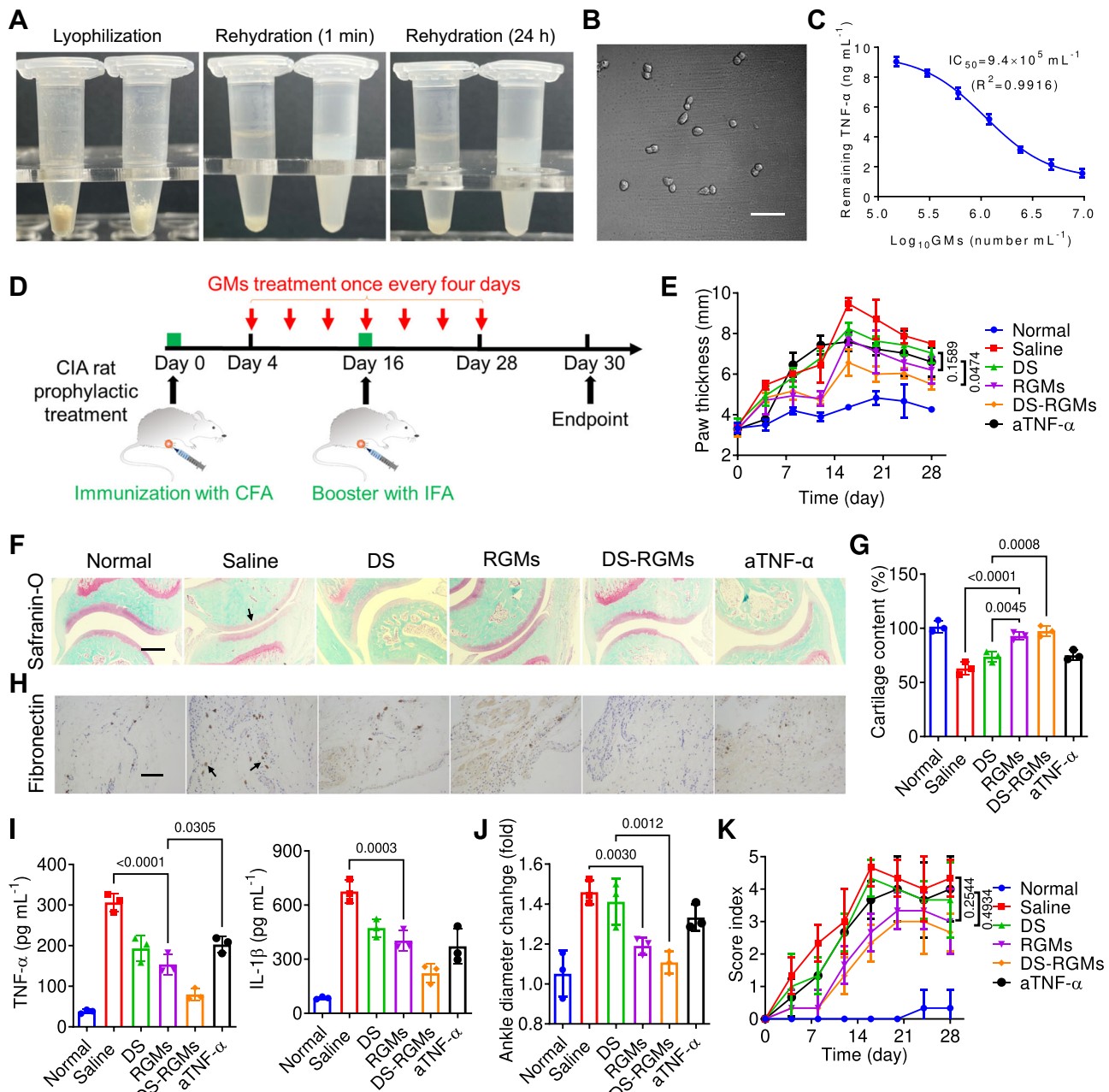

**Fig. 6 | RGMs prevented the joint injuries in CIA rats. A** Lyophilized form of GMs. Lyophilized MAs (Non-gelated MAs) served as a control. After storage in room temperature for 1 month, the powers were rehydrated in PBS for 1 min and 24 h. **B** The morphology of RGMs under microscopy. Scale bar: 50 µm. **C** Different doses of RGMs (0.1875, 0.375, 0.75, 1.5 and $3 \times 10^6\,mL^{-1}$) were incubated in PBS solution containing $10\,\mu g\,mL^{-1}$ of TNF-α for 1 h, and the supernatant concentration of TNF-α was measured by Elisa kits. **D** Scheme of phylactic treatment on CIA rats. **E** The changes of paw thickness in CIA rats treated with saline, DS, RGMs, DS-RGMs and aTNF-α. **F, G** Safranin-O staining on the ankle sections from treated rats (**F**) and cartilage content quantification (**G**). Scale bar: 200 µm. **H** Fibronectin staining on the knee sections from treated rats. Scale bar: 100 µm. **I** The serums from treated rats were collected on Day 28, and the concentrations of TNF-α and IL-1β were measured by Elisa kit. **J** The ankle diameter of treated rats recorded on Day 28. **K** The changes of arthritis score in treated rats recorded every 4 days with a total of 28 days. Representative photos in (**F**–**H**) came from independent experiments on three different ankles and synovial intimal linings, respectively ($n = 3$). All data was presented as mean ± s.d. ($n = 3$). All statistical analyses were conducted by using One-Way ANOVA.

respectively at the same fluorescence intensity for ex vivo fluorescent imaging (Supplementary Fig. 9A, B), which referred to as $5 \times 10^6$ of DiD-GMs and DiD-EGMs, and $2 \times 10^7$ of DiD-GRBCs, respectively. After injection for 6 h, both DiD-GRBCs and DiD-EGMs treated mice exhibited nearly no fluorescence intensity in the lung tissue, and they were mainly distributed in the liver. In contrast, a bright fluorescent intensity was observed in the DiD-GMs treated mice, further suggesting that GMs exhibited more significant inflammatory tropism than GRBCs and EGMs. This result indicated that macrophage membrane protein

played a key role in the targeted delivery to inflammatory lung. To further verify the in vivo inflammatory tropism of GMs, *i.v.* injection of FITC labelled GMs (via FITC conjugation with hydrogel) exhibited a large number of single green fluorescence dots in the lung sections, whereas weak fluorescence was detected in mice treated with free FITC, or the mixture of FITC and GMs (FITC+GMs) (Fig. 7C). Furthermore, FITC labelled GMs were stained with a membrane dye (DiI) in order to measure the spatial localization of DiI stained membrane and intracellular FITC conjugated hydrogel. As shown in Fig. 7D, the green

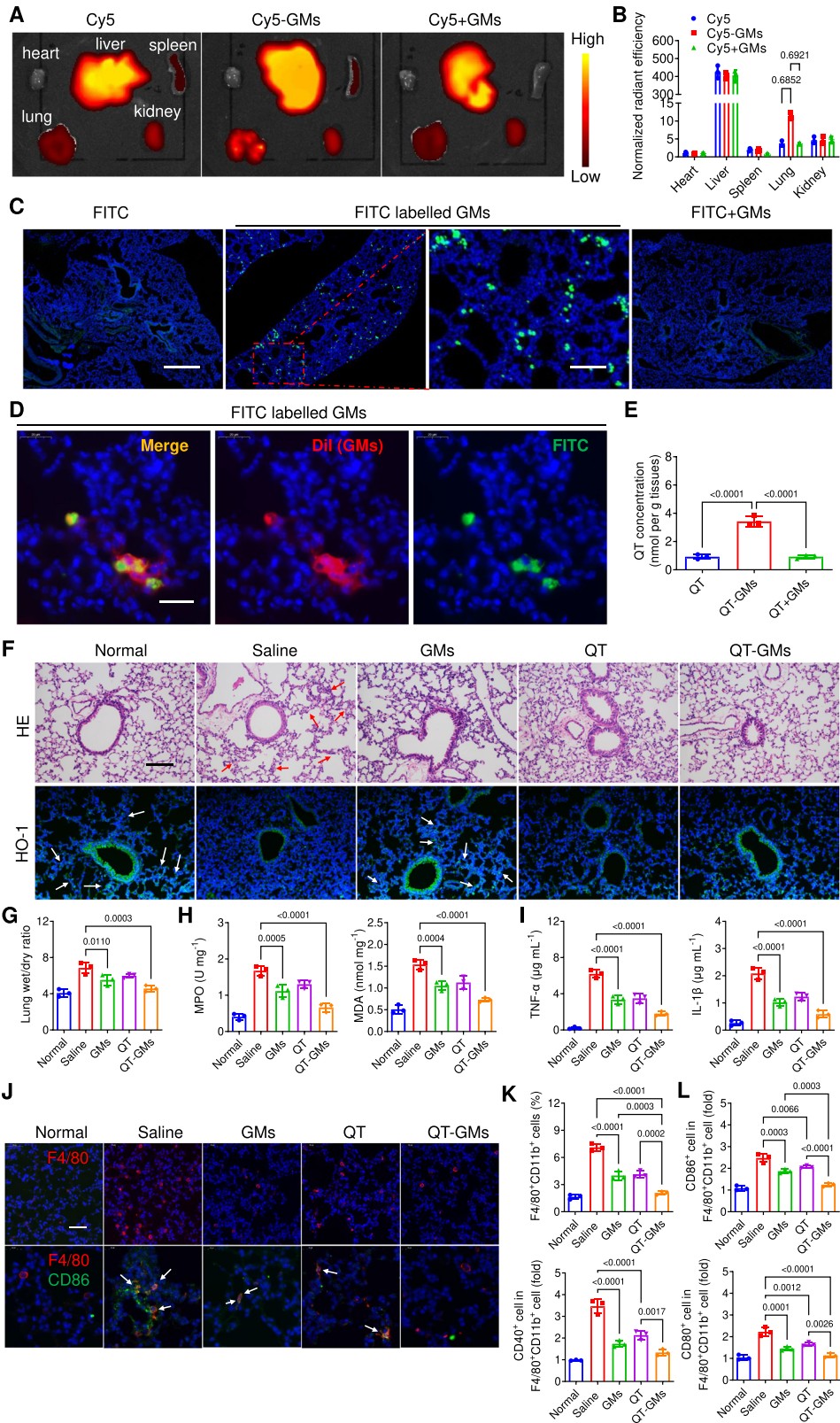

fluorescence of intracellular FITC was surrounded by red fluorescent membrane of GMs in the inflammatory lungs' tissue sections. Collectively, these results show that GMs may efficiently accumulate in the inflammatory lungs, thereby enhancing the targeting rate of intracellular payload in lungs of AP mice.

To conduct in vivo pharmacokinetic study on GMs, CytoTrace Red CFDA, a red fluorescent cell staining dye, was selected to label GMs via

reactions with cellular components to form cell-impermeant products. AP mice were i.v. injected with $5 \times 10^6$ of Red CFDA-labelled GMs ($n = 3$), and the organs (heart, liver, spleen, lungs and kidneys) were collected for ex vivo fluorescent imaging after administration for different durations (1, 6, 12, 24, 48 and 72 h). All mice showed bright red fluorescence in the lungs immediately after administration, and the fluorescence intensity quickly decreased but still maintained at a

**Fig. 7 | Targeted therapy of acute lung inflammation in AP mice by GMs. A**, **B** AP mice were *i.v.* administered with Cy5, Cy5-GMs, Cy5+GMs at 0.5 mg kg⁻¹ of Cy5 and 5 × 10⁶ of GMs, respectively (*n* = 3). After injection for 10 min, different organs (heart, liver, spleen, lungs and kidneys) were collected for ex vivo fluorescence imaging (**A**) and the fluorescence intensity was quantified by IVIS (**B**). **C** AP mice were *i.v.* administered with FITC, FITC labelled GMs, FITC+GMs at 0.1 mg kg⁻¹ of FITC, respectively. After injection for 10 min, the lung tissues were sectioned for fluorescence imaging. Scale bar: 500 μm. Amplified scale bar: 125 μm. **D** FITC labelled GMs (with green fluorescence cytosol) were stained with DiI. After administration of the dye labelled GMs for 10 min, the lung tissues of AP mice were sectioned for fluorescence imaging. Scale bar: 20 μm. **E** After *i.v.* administration with saline, QT, GMs and QT-GMs at 3 × 10⁶ of GMs per mice and 5 mg kg⁻¹ of QT for 10 min, QT concentration in the lungs of AP mice was quantified by HPLC. **F** AP mice were *i.v.* administered with saline, QT, GMs and QT-GMs at 3 × 10⁶ of GMs per mice and 5 mg kg⁻¹ of QT, respectively. After administration for 6 h, the lung tissues were collected for HE staining and HO-1 staining. Scale bar: 100 μm. **G** The wet/dry ratio of the collected lung tissues. **H** Serum levels of TNF-α and IL-1β. **I** The concentrations of MPO and MDA in the lung tissues were analyzed by assay kits. **J** Fluorescence imaging on the infiltration of F4/80⁺ cells in the lungs of treated mice. Scale bar: 50 μm. **K**, **L** Flow cytometry analysis on the infiltration of F4/80⁺CD11b⁺ cells and the ratio of CD86⁺, CD40⁺, and CD80⁺ cells among them in the lungs of treated mice. All data was presented as mean ± s.d. (*n* = 3). Representative photos in (**C**, **D**–**F**–**J**) came from independent experiments on three different lungs (*n* = 3). Statistical analysis of (**B**) was conducted using Two-Way ANOVA. All the other statistical analyses were conducted using One-Way ANOVA.

certain level for 48 h (Supplementary Fig. 9C). After administration for 6 h, the spleen started to show fluorescence signal and quickly increased, but eventually disappeared post administration for 48 h. The liver intensity followed a similar pattern of the fluorescence changes of the spleen, and continuously increased in 48 h post administration but eventually decreased at 72 h. Together with a large area under fluorescence curve (AUC) in the lungs, spleen and liver (Supplementary Fig. 9D), these results indicated that GMs had a good response to accumulate in the inflammatory lung tissue, and then may get phagocyted by monocyte phagocytic system in the spleen to break down. Finally, GMs debris that could not be reutilized was transferred to the liver, and metabolized to produce water-soluble substances and excreted with urine, or directly excreted with faces to the intestinal tract. Blood was collected from the other batch of mice at pre-determined timepoints (0.1, 0.5, 1, 2, 3, 4.5, 6, 12, and 18 h). As shown in Supplementary Fig. 9E, the blood fluorescent possessed highest intensity after administration for 0.1 h, and quickly decreased at the beginning. Then the fluorescence signal slowly decreased and almost disappeared after administration for 18 h. According to the fluorescence intensity-time curve (Supplementary Fig. 9F), GMs exhibited a long half-life of drug clearance ($t_{1/2} = 5.6$ h), indicating sufficient circulation time for GMs to neutralize systemic proinflammatory cytokines, due to the intact cell morphology and membrane structure of GMs to avoid fast clearance by monocyte phagocytic system.

Quercetin (QT), a natural anti-oxidation compound, was loaded into GMs during the intracellular gelation step to yield QT-GMs, for potentially combined treatment of AP in mice. AP mice were blindly and randomly divided into three groups (*n* = 3), and *i.v.* administered with QT, QT-GMs, and mixture of QT and GMs (QT+GMs) at 3 × 10⁶ of GMs per mice and 5 mg kg⁻¹ of QT. After administration for 10 min, QT concentration in the lung tissue was much higher in the QT-GMs treated mice (3.42 nmol per kg tissue), in comparison to that of QT treated mice (0.93 nmol per kg tissue) and that of QT+GMs treated mice (0.92 nmol per kg tissue) (Fig. 7E). HE staining showed that LPS induced severe lung injuries including infiltration of immune cells in lung interstitium and alveolar space, and thickening of alveolar wall. Hemeoxygenase-1 (HO-1) expression contributed to the defence against oxidative and inflammatory insults in the lung[42]. Therefore, HE staining and HO-1 immunohistochemical analysis was conducted on the lung sections after administration for 6 h, which showed that GMs efficiently inhibited the infiltration of immune cells and improved HO-1 expression (Fig. 7F). QT moderately attenuated the lung inflammation, and significantly improved therapeutic efficacy was observed when it was delivered by GMs. Moreover, GMs obviously decreased the wet weight/dry weight ratio (wet/dry ratio) of the lung tissue (Fig. 7G), and inhibited the expression of inflammatory cytokines (TNF-α and IL-1β), comparable to QT treatment (Fig. 7H). Myeloperoxidase (MPO) and malondialdehyde (MDA) play a key role in the neutrophil infiltration and oxidative progression in the inflammatory lung diseases[43]. Both markers were obviously decreased by GMs treatment, similar to QT

group, and QT-GMs jointly reduced the level of both markers in a more pronounced manner than QT or GMs alone (Fig. 7I). Meanwhile, immunofluorescence staining of the lung section showed that LPS treatment induced significant infiltration of F4/80⁺ cells (one marker of macrophage) and F4/80⁺CD86⁺ cells (Fig. 7J). Flow cytometry analysis further quantified the increased percent of CD86⁺, CD40⁺ and CD80⁺ cells, respectively, in F4/80⁺CD11b⁺ cells (Fig. 7K, L and Supplementary Fig. 10), confirming that LPS treatment locally polarized macrophage into proinflammatory phenotype. Due to the neutralization effect towards proinflammatory cytokine and bacteria product (LPS), GMs treatment reduced the infiltration ratio of proinflammatory macrophages in pneumonia mice. In addition, QT was reported to induce inhibitory effects on inflammatory macrophage polarization via the NF-κB and IRF5 signaling[44], which was also confirmed by the decreased percentages of CD86⁺, CD40⁺ and CD80⁺ cells, respectively, in F4/80⁺CD11b⁺ cells in the lungs (Fig. 7K, L and Supplementary Fig. 10). Subsequently, the combination therapy of QT-GMs exhibited the most pronounced therapeutic outcome in decreasing F4/80⁺CD11b⁺ cells infiltration and efficiently inhibiting macrophage polarization into proinflammatory phenotype, when compared with other therapeutic groups. The effect of QT-GMs treatment on macrophage population was mainly beneficial from cytokine or LPS neutralization effect of GMs and anti-inflammation effect of QT. AP mice treated with 2 mg Kg⁻¹ of hydrogel, 3 × 10⁶ of MAs, and 3 × 10⁶ of GRBCs served as control groups to verify the therapeutic efficacy of QT-GMs. As a result, MAs, hydrogel, GRBCs and EGMs showed negligible anti-inflammation effects, and MAs treatment even slightly aggravated lung tissue inflammation (Supplementary Fig. 11), indicating the important role of membrane inflammation receptor protein in neutralization treatment. Immunofluorescent staining of neutrophil cells (CD11b⁺Gr-1⁺ cells) was further conducted in the lung tissues of treated AP mice. As shown in Supplementary Fig. 12, the fluorescent intensity of CD11b⁺Gr-1⁺ cell was strong in AP mice treated with MAs (-16.2%) and hydrogel (-15%), comparable to saline treated mice (-15.8%). GMs and QT both decreased the fluorescent neutrophils in the lung sections, and a significant reduction of neutrophil infiltration was detected in QT-GMs treated mice (-4.7%), attributed to the combined anti-inflammation effect of QT and GMS. There results confirmed that QT-GMs effectively ameliorate the acute lung inflammation via both cytokine neutralization of GMs and anti-inflammatory effects of QT. Considering that GMs is rigid and does not have the plasticity of living macrophages, it is hard to penetrate through lung vasculature, and most of them are located inside the abundant lung capillaries. Thus, GMs neutralize cytokines in the lungs, capillary released from inflammatory pneumonocyte. In addition to lungs, GMs were mainly distributed in the liver. When blood flew through these tissues, GMs sequestrated the cytokines in the blood and exhibited the systemic anti-inflammation effect. Moreover, the accumulated drug release ratio from GMs reached ca. 65% after incubation in PBS for 6 h (Fig. 2G). After accumulation in the inflammatory lungs, the shear stress in abundant lung capillaries could push QT-GMs to release the intracellular QT more quickly, and then the

drug went through lung capillaries and penetrated into inflammatory pneumonocyte for anti-inflammation therapy.

### In vivo safety evaluation of GMs

GMs were constructed from natural MAs and intracellular hydrogel derived from Phe-CS and CB[8], and both components had a high biocompatibility[45,46]. For potential clinical translation, the safety of GMs was systemically evaluated in SD rats. $5 \times 10^6$ of GMs was *i.v.* injected into SD rats once a day for 3 days, and the blood serum and different organs (heart, liver, spleen, lungs and kidneys) were collected after administration for 3 days. HE staining on the section of different organs showed normal pathological conditions (Supplementary Fig. 13A). The number of white blood cells (WBC), red blood cells (RBC) and platelets (PLT) did not change in comparison to that of the control group (Supplementary Fig. 13B), and the serum cytokines (TNF-α and IL-1β) remained at a normal level (Supplementary Fig. 13C). Finally, the levels of the liver and kidney function markers were similar to those of the control group, including alanine aminotransferase (ALT) and aspartate transaminase (AST) (Supplementary Fig. 13D), and blood urea nitrogen (BUN) and uric acid (UA) (Supplementary Fig. 13E). These results suggested a good safety profile of GMs, and with significant translational potential. Nevertheless, because a large number of GMs were transferred to the liver for metabolization, the potential liver autoimmune response was further evaluated. As shown in Supplementary Fig. 14A, the infiltration of CD4+ T and CD8+ T cells in mouse liver was measured to investigate the T-cell-mediated autoimmune response. Nearly no infiltration of CD4+ and CD8+ cells was observed in the liver from mice treated with GMs for 7 days and 14 days, respectively. At the same time, the mouse liver showed a similar appearance to the liver of healthy mice as well, without any symptoms of inflammation (Supplementary Fig. 14B). HE staining analysis showed no infiltration of inflammatory cells (Supplementary Fig. 14C), and Sirius red staining on the liver exhibited negligible fibrosis level after GMs treatment (Supplementary Fig. 14D), which was otherwise the main feature of terminal autoimmune hepatitis. Finally, immunofluorescence analysis of α-smooth muscle actin (α-SMA) was further conducted on the mouse liver (Supplementary Fig. 14E), and the results showed that GMs treatment did not increase α-SMA expression around the blood vessels, which was otherwise the other evidence of fibrosis in autoimmune hepatitis mouse. Together with the result of normal histological feature (Supplementary Fig. 13), GM treatment did not induce noticeable liver autoimmune response and liver toxicity, attributed to the quick clearance and quite low accumulation in the liver after *i.v.* treatment for 3 days (Supplementary Fig. 9C, D).

## Discussion

This study validated an intracellular supramolecular gelation strategy, and accordingly we developed GMs for targeted treatment of various inflammatory diseases. This strategy inherited the strengths, and avoided the weaknesses, of both immune cell mimetic nanomedicines and immune cell-based carriers, affording "inert" macrophages (GMs) with completely intact cell membranes for efficient neutralization towards multiple inflammatory cytokines. In addition, GMs can have drugs loaded into the cytoplasmic hydrogel with a high drug loading efficiency and a slow release kinetics, allowing targeted drug delivery to the inflammatory tissues, attributed to the inflammatory tropism of GMs, for combined inflammation modulation. Importantly, GMs can be facilely prepared, lyophilized and stored at room temperature for at least a month, making the product readily available for remote areas as well as for facilitating potential clinical translation.

In particular, GMs are constructed via intracellular infusion of Phe-CS after a freeze-thaw cycle and subsequent infusion of CB[8] for intracellular crosslinking and gelation. GMs show an intact cellular structure identical to the source MAs, and exhibit slow drug release behaviour as well as no inflammation activation ability. In vitro cell co-culture and dynamic conjugation on microfluidic chip show that GMs may selectively bind with inflammatory cells in static condition and under flow condition. Due to the receptor-ligand interactions between GMs and inflammatory cytokines and cells, GMs exhibit extended retention in the inflammatory paw of RA rats, and may serve as inhibitor of multiple cytokines to efficiently alleviate arthritis pathology, which is further strengthened by intracellularly loaded medicine. Lyophilized GMs allow longtime storage at room temperature, and RGMs effectively prevent the RA progression. Moreover, in response to inflammatory signal of acute pneumonia, GMs may act as a drug carrier for targeted delivery of intracellular medicine to inflammatory tissues such as the inflammatory lung tissues of AP mice, which is further confirmed by the co-localization of DiI stained membrane of GMs and FITC conjugated cytoplasm hydrogel of GMs in the lung sections. Beneficial from cytokine neutralization of GMs and anti-inflammatory effect of intracellular medicine (QT), QT-GMs jointly ameliorate acute pneumonia. Because bone-marrow derived macrophage and monocyte derived macrophage have a wide range of cytokine and chemokine receptors, therefore are more prone to respond to body cytokines and chemokines. Accordingly, GMs prepared from those macrophages will likely have better sequestration powers to multiple inflammatory factors and pathological molecules.

Due to the receptor-ligand interaction, cell membrane camouflaged nanomedicine has developed rapidly in recent years, and they could also absorb and neutralize multiple pathological molecules[47,48]. However, membrane protein disintegration, loss and spatial disorder are inevitable problems during the construction of membrane coated nanomedicine, which reduces their neutralization efficiency. Thus, one of the highlights of this work is that the gelated cell highly preserved intact cell membrane structures, membrane protein mobility and membrane lipid order. Gelated cell was more effective to neutralized cytokines and chemokines in comparison to membrane coated nanomedicine under the same number of source cells. Secondly, intracellular hydrogel could be utilized for drug loading, and gelated cell could act as cell-based drug carrier for hitchhiking delivery. This design has addressed the main challenges of drug metabolization and efflux in the current living cell-based carriers. Finally, clinical monoclonal antibody only inhibits one or a few of pathological molecules, and most of them are stored at low temperatures. Gelated cell inherits intact protein components of source cell and shows a high neutralization efficiency towards multiple pathological molecules. Hydrogel cytoskeleton has endowed gelated cell with capability for freeze drying treatment that allows storage and transportation at a room temperature, which will benefit potential clinical translation.

Nevertheless, there are several limitations in our study regarding its potential clinical translation. First, although GMs exhibit a decent safety profile in rats, the hydrogel derived from Phe-CS and CB[8] are not approved by FDA. Second, intracellular gelation was conducted with MAs from rats and mice, however it is often difficult to separate a large number of human MAs, which may limit its clinical translation. Therefore, other immune cells such as neutrophils could be utilized during the clinical application, due to the same surface receptors and cytokines interactions and inflammatory tropism effects, and this strategy could be in principle extended to all other types of cells for a variety of other potential biomedical applications.

## Methods
### Materials

Phenylalanine, chitosan (average molecular weight of 10,000 Da), quercetin, diclofenac sodium, lipopolysaccharide, complete Freund's adjuvant, and incomplete Freund's adjuvant were purchased from Sigma (USA). All other chemicals used in this study were got from Aladdin (China). Curcubit[8]uril was prepared in laboratory according to previous reported method. Cell membrane dye DiI and DiO were purchased from Beyotime (China). PerCP/Cy5.5-CD86 antibody

(Cat No. 105027), APC-CD206 antibody (Cat. No. 141707), PE-F4/80 antibody (Cat. No. 157303), FITC-CD11b antibody (Cat. No. 101205) were obtained from Biolegend (USA). TNFR1 antibody (Cat. No. 21574-1-AP), TNFR2 antibody (Cat. No. 19272-1-AP), IL-6Rα antibody (Cat. No. 23457-1-AP), IL-6Rβ antibody (Cat. No. 67766-1-Ig), IL-1R2 antibody (Cat. No. 60262-1-Ig), gp130 antibody (21175-1-AP), CD248 antibody (Cat. No. 60170-1-1 g), and fibronectin antibody (Cat. No. 66042-1-1 g) (Supplementary Table 1) were purchased from Proteintech (China).

RAW264.7 cells, HUVECs, L-O2 cells, and RLE-6TN cells were obtained from American Type Culture Collection (ATCC, USA), and verified by mycoplasma detection, isozyme detection, and DNA fingerprinting. Foetal Gibco Dulbecco's modified Eagle's medium (DMEM), Foetal Bovine Serum (FBS) and Trypsin-EDTA (0.25%) were obtained from Gibco (China). SD rat (stock number: 101) and C57BL/6 mouse (stock number: 219) were collected from Guangdong Vital River Laboratory Animal Technology Co., Ltd. The type of animal facility was specific pathogen free (SPF) level, and experimental/control animals were bred separately. All animal experiments were approved by the Animal Ethics Committee, China Technology Industry Holdings (Shenzhen) Co., Ltd, and were conducted in accordance with the Animal Management Rules of the Ministry of Health of the P. R. China.

### Synthesis of Phe-CS
50 mg of chitosan (0.31 mM monosaccharide) was firstly dissolved into 25 mL of acetate buffer (pH 5.5), and 56.3 mg of Boc-Phe-OSu (0.155 mM) in 100 µL of DMF was dropwise added into chitosan solution. The mixture was stirred for 24 h at 50 °C under dark, and the unreacted substrates were removed via dialysis against pure water (MWCO 3500) at the end of reaction. The final products were obtained via lyophilization, and the structure of Phe-CS was confirmed via $^1$H NMR in $D_2O$.

### Preparation and characterization of CS hydrogel
To prepare supramolecular CS hydrogels, Phe-CS was firstly dissolved in PBS (pH 5.0), and CB[8] in PBS was added into Phe-CS under continuous stirring and sonication until a cloudy and viscoelastic hydrogels formed. To evaluate the influence of the Phe-CS concentration on the properties of hydrogels, an increased amount of Phe-CS (0.1, 0.5, 1.0, 2.0, and 5.0, wt%) was respectively dissolved in acidic buffer, and corresponding CB[8] was added into Phe-CS to yield hydrogels. The rheologic properties of hydrogels were tested on a rheometer (DHR-1, TA Instruments) equipped with a temperature controller. To obtain SEM images of hydrogels, the fabricated hydrogel was lyophilized and fixed on silicon slice. The hydrogel was sprayed with gold and imaged under SEM at 2.0 kV.

### Preparation and characterization of GMs
Macrophage was incubated with 2% (wt%) of Phe-CS solution, and the culture dish was placed in a −80 °C refrigerator for 15 min. Then, the dish was taken out and thawed in a 37 °C water bath. After quick rinse with PBS solution to remove residual Phe-CS on cell surface, 50 µM of CD[8] was added to the cell culture dish, and incubated at room temperature for 5 min. Finally, the cell medium was quickly raised by PBS solution again, and GMs was obtained and stored at −80 °C for further experiments. The morphology of GMs was imaged by SEM, and fluorescence imaging was conducted on DiI stained GMs prepared from FITC conjugated hydrogel. The diameter of GMs was measured by DLS and analyzed using Zetasizer Software (version 7.11). In order to construct the drug-loaded GMs, the drug can be mixed with Phe-CS solution, followed by above preparation method of GMs. The drug loaded efficiency and drug release ratio in PBS was quantified by HPLC. GMs was stored in −80 °C refrigerator for further usage. The drug loading capacity of DS-GMs used for anti-arthritis treatment was ~1.4%. In the treatment of acute pneumonia, the weight content of QT in

QT-GMs was ~4.2%. Graphpad Prism 10 was used for statistical analysis and data plotting.

### Measurement of membrane protein
Membrane protein of GMs was collected by using membrane protein extraction kit, and sample protein concentration was quantified by BCA kit. To detect the protein type, the sample protein was performed with gel electrophoresis, and then stained with Coomassie blue dye. After de-staining at room temperature and soaking in deionized water, the gel was imaged to obtain protein bands. To western blot analysis, the PVDF was utilized to cover the gel after electrophoresis. Then, the PVDF membrane was incubated with primary antibodies of IL-6Rα, IL-6Rβ, TNFR1, TNFR2 and IL-1R2 at 4 °C overnight, and incubated with the HRP-conjugated anti-rabbit antibody and HRP-conjugated anti-mouse IgG antibody at room temperature for 1 h. Finally, the PVDF membrane was rinsed with pure water for 5 times, and the chemiluminescence and imaging were conducted.

### Measurement on membrane protein fluidity
As a classical signaling pathway, IL-6R receptor of cell membrane binds with IL-6 ligands, and then the gp130 proteins of cell membrane would bind with IL-6/IL-6R complex to initiate intracellular signal transduction. Therefore, in order to verify the membrane protein fluidity of GMs, the IL-6R and gp130 of cell membrane were labelled with IL-6R antibody (green light) and gp130 antibody (red light). After washing with PBS for 3 times, IL-6 was added and incubated for 12 h. Then, the distribution of cell membrane proteins (IL-6R and gp130) was imaged by confocal fluorescence microscopy using LAS X software (version 3.5.2). Fixed macrophages with 4% PFA served as control group.

### Measurement on membrane lipid order
GMs were incubated in medium containing 100 µg ml$^{-1}$ of Laurdan dye for 1 h. After washed with PBS for 3 times, fluorescent images were obtained by confocal microscopy for lipid order analysis of cell membrane. Specifically, the mean excitation wavelength was 405 nm, and emission wavelength of 440−460 nm referred to ordered membrane lipid and emission wavelength of 490−510 nm referred to disorder membrane lipid. All images were exported in TIFF format, and ImageJ was used to calculate the generalized polarization value (GP value) of these fluorescence images and create a pseudo-coloured GP intensity merged image and intensity histogram. The GP value was calculated according to the formula of GP value = $(I_{440-460\ nm} - I_{490-510\ nm})/(I_{440-460\ nm} + I_{490-510\ nm})$, where I is the pixel intensity. Finally, the GP distribution of GMs was analyzed, and free macrophages and PFA-fixed macrophages served as control group.

### In vitro sequestration of GMs towards inflammatory cytokines
10 uM of both TNF-α and IL-1β solution was added with different doses of GMs (0.15, 0.3, 0.6, 1.2, 2.4, 4.8 and 9.6 × 10⁶ mL$^{-1}$), and incubated for 1 h. Then, the supernatant was collected by centrifugation, and the concentrations of TNF-α and IL-1β were measured by Elisa kit and the data was collected using a plate reader using SoftMax Pro 5.4.1. The precipitated cell was fluorescently labelled with F4/80 antibody and CD86 antibody, and the proportion of F4/80⁺CD86⁺ cells was analyzed by flow cytometry. In order to study the dose-dependence manner, different qualities of GMs were added to 10 uM of both TNF-α and IL-1β solution, and the cytokines in the supernatant was measured after incubation for 2 h. The correlation curve between the mass of GMs and the concentration of inflammatory cytokines in the solution was drawn finally.

### Static binding between GMs and inflammatory cells
Human umbilical vein endothelial cells (HUVECs) were treated with 10 µg mL$^{-1}$ of LPS to induce an inflammatory synovial cell model. Then GMs were added and incubated with inflammatory HUVECs for 12 h.

After wash with PBS, the binding result between GMs and inflammatory HUVECs was observed under confocal microscope. Furthermore, to analyze the binding rate, GMs and HUVECs were stained with DiO and DiI, respectively. The proportion of $DiO^+DiI^+$ cells was quantified by flow cytometry. The data was collected by BD Accuri C6 Software (version 1.0.264.21), and analyzed using FlowJo software (version 7.6.1).

### Dynamic binding between GMs and inflammatory cells

In order to study the inflammatory tropism of GMs under blood flow conditions, a single-channel microfluidic chip was constructed. Rat synovial cells (RSC-364 cells) were incubated in the chip, and $10 \, \mu g \, mL^{-1}$ of LPS was added to induce inflammatory synovial cells. After injection with a certain number of FITC conjugated GMs, the dynamic binding with inflammatory synovial cells and GMs in chip was observed under fluorescent microscope at different time points (1 min and 2 min). Moreover, a dual-channel microfluidic chip was further constructed, normal synovial cells were cultured in the upper channel and inflammatory synovial cells were cultured in the lower channel. After injection with DiO labelled GMs, PFA-MAs and Nongelated MAs at a flow rate of $200 \, \mu L \, min^{-1}$, confocal fluorescence imaging was performed at 2 min. The fluorescence intensity was quantified by ImageJ 1.8.0.345.

### Construction of RA rat and retention effect of GMs in ankle joint

Six-week-old female SD rats were intradermally injected with $100 \, \mu L$ of complete Freund's adjuvant at the footpad of a rear paw. After administration for 14 days, $100 \, \mu L$ of incomplete Freund's adjuvant was injected to boost inflammation. The inflammatory response of the joints reached a peak at day 28, and RA mice was constructed. Cy5 served a fluorescence model drug and loaded in GMs to study the drug retention in ankle joint. RA mice were randomly divided into three groups ($n = 3$), and in situ administered with free Cy5, Cy5-GMs and mixture of Cy5 and GMs (Cy5+GMs) at a dose of $0.5 \, mg \, kg^{-1}$ Cy5 and the same number of GMs, respectively. In vivo imaging was conducted in the swelling paw after administration for different times (0 h, 12 h, 24 h, 36 h, 48 h, 60 h, 72 h) using IVIS (Lumina XR III), and the fluorescence intensity was recorded.

### Prophylactic and therapeutic treatment of GMs on RA rat

In the phylactic treatment of arthritis, 6-week-old female RA rats were randomly divided into 5 groups ($n = 6$) after the injection of complete Freund's adjuvant at day 0, and in situ administered with saline, DS, GMs and DS-GMs at $3 \times 10^6$ of GMs per rat and $1 \, mg \, kg^{-1}$ of DS once every 4 days until Day 28. Rat treated with $1 \, mg \, kg^{-1}$ of anti-TNF-α (aTNF-α), and normal rat served as positive control and normal control. $100 \, \mu L$ of incomplete Freund's adjuvant was intradermally injected at Day 14. The morphology and thickness of rat paw were imaged and recorded once every 5 days until Day 28. At the endpoint of experiment, the ankle diameter was measured, and ankle joint and serum were collected from rats euthanatized with $CO_2$. Serum inflammatory levels (TNF-α and IL-1β) were determined by Elisa kit, and pathological section analysis of ankle joints (HE staining and Safranin O staining) was carried out. Immunohistochemical staining of CD248 were performed on synovial tissue. Finally, disease activity scores of RA were conducted to evaluate the swelling and redness of rat paws. In the therapeutic treatment of arthritis, RA rats were randomly divided into 5 groups ($n = 6$) after RA model was constructed at Day 28, and in situ administered with saline, DS, GMs, DS-GMs, aTNF-α, and DS+ aTNF-α at $3 \times 10^6$ of GMs per rat, $1 \, mg \, kg^{-1}$ of DS and $1 \, mg \, kg^{-1}$ of aTNF-α once every 4 days until Day 52. Rats treated with $2 \, mg \, kg^{-1}$ of hydrogel, $3 \times 10^6$ of MAs and $2 \, mg \, kg^{-1}$ of PSL, and MM-NPs prepared from $3 \times 10^6$ of MAs served as control groups. The follow-up efficacy evaluation was carried out according to the previous evaluation methodology of phylactic treatment.

### Construction of AP mouse and targeted delivery of GMs

Six-week-old C57BL/6 male mice was administered with $50 \, \mu L$ of LPS ($1 \, mg \, mL^{-1}$) solution via intranasal route. After treatment for 1 h, the phenotype of acute pneumonia appeared, and AP mice was constructed. Cy5 served a fluorescence model drug and loaded in GMs to investigate the drug biodistribution. AP mice were randomly divided into three groups ($n = 3$), and i.v. administered with free Cy5, Cy5-GMs and mixture of Cy5 and GMs (Cy5+GMs) at $0.5 \, mg \, kg^{-1}$ of Cy5 and $5 \times 10^6$ of GMs, respectively. After injection for 10 min, different organs (heart, liver, spleen, lung and kidney) were collected for ex vivo fluorescence imaging. Furthermore, AP mice were randomly separated into 3 groups ($n = 3$), and i.v. administered with free FITC, FITC labelled GMs and FITC+GMs at a dose of $0.1 \, mg \, kg^{-1}$ FITC and the same number of GMs, respectively. After injection for 10 min, lung tissues were collected and sectioned for immunofluorescent imaging. Moreover, the membrane of GMs was labelled with DiI (red fluorescent dye), and FITC was conjugated onto Phe-CS to label the intracellular hydrogel of GMs. The biodistribution of GM membrane and intracellular hydrogel in the lung section was imaged by IVIS.

### Therapeutic treatment of GMs on AP mouse

AP mice were i.v. administered with saline, QT, GMs, and QT-GMs at $5 \, mg \, kg^{-1}$ of QT and $3 \times 10^6$ of GMs. After administration for 6 h, the mice were euthanized and QT concentration in lung tissue was detected by HPLC. In addition, lung tissues were sectioned and conducted for HE staining and HO-1 staining. Immunofluorescent staining of macrophages ($F4/80^+$ cells) and proinflammatory type ($F4/80^+CD86^+$ cells) was performed in lung tissue, and the ratios of macrophages and proinflammatory type were quantified by flow cytometry. Furthermore, dry weight/wet weight ratio of lung tissue was recorded, and the concentrations of MDA and MPO in lung tissue was analyzed by detection kits. Inflammatory factor levels (TNF-α and IL-1β) in serum were determined by Elisa kits.

### In vivo safety evaluation of GMs in rat

In vivo safety of GMs was evaluated in SD rat. After i.v. administration with GMs at $5 \times 10^6$ once each day for total 3 days, the serum and different organs (heart, liver, spleen, lung and kidney) were collected. Serum TNF-α and IL-1β were analyzed by Elisa kit. Liver function markers (serum ALT and AST) and renal function markers (serum BUN and UA) in serum were quantified by detection kits. HE staining were conducted on heart, liver, spleen, lung and kidney to analyze histological damage. The number of blood immune cells was quantified by cell counter.

### Statistical analysis

One-way ANOVA and Two-way ANOVA were utilized for statistical analysis, and value of $*P \leq 0.05$, $**P \leq 0.01$ and $***P \leq 0.001$ were applied to annotate statistical significance. All in vitro experiment were performed independently for at three times, and animal size for in vivo experiment was set as $n = 6$. All data were presented as mean value ± the standard deviation.

### Reporting summary

Further information on research design is available in the Nature Portfolio Reporting Summary linked to this article.

## Data availability

All other data are available in the article and its Supplementary files or from the corresponding author upon request. Source data are provided with this paper.

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

## Acknowledgements

We thank the public platform of State Key Laboratory of Quality Research in Chinese Medicine (University of Macau) for providing the fundamental experimental facility. This work was financially supported by the Science and Technology Development Fund (FDCT) of Macau SAR (0086/2022/A2, 0065/2021/A2, 0001/2023/RIA1, 0070/2023/RIA2, 0006/2022/ITP and SKL-QRCM(UM)-2023-2025), National Natural Science Foundation of China (22071275, 22271323, 32001016 and 32371381), Shenzhen Science and Technology Innovation Commission (EF023/ICMS-WRB/2022/SZSTIC and SGDX20210823103803027), University of Macau—Dr. Stanley Ho Medical Development Foundation "Set Sail for New Horizons, Create the Future" Grant 2023 (SHMDF-VSEP/2022/001), University of Macau (MYRG-CRG2022-00011-ICMS), and Ministry of Education Frontiers Science Centre for Precision Oncology, University of Macau (SP2023-00001-FSCPO).

## Author contributions

The project was conceptually designed by R.W. and C.G. The majority of the experiments were conducted by C.G., Q.W. and Y.D. (with equal contributions), assisted by B.X. and J.L. Data analysis and interpretation was done by J.W. and C.K., assisted by S.L. and G.M. The manuscript was prepared by C.G. and R.W. All authors discussed the results and implications, and commented on the manuscript.

## Competing interests

The authors declare no competing interests.
