## [Peer Review File · Nature Communications]

Targeted therapies of murine inflammatory diseases with intracellularly gelated macrophagesREVIEWER COMMENTS

Reviewer #1 (Remarks to the Author):

In this paper, the authors developed an intracellularly gelated macrophages (GMs) with highly preserved cell membrane structures, which can function as a natural antiinflammation agent and stable drug carrier to treat inflammatory diseases of rheumatoid arthritis and acute pneumonia. Although an interesting work, there are a few scientific shortcomings. It needs to carry out some additional experiments to confirm some of the results in this work.

1. Please highlight the novelty by comparing with previous works.
2. For in vivo experiments, the MAs and hydrogel are better to be used as control.
3. The DS, QT and hydrogel loading contents of GMs should be provided.
4. The swelling properties of hydrogels require further evaluation.
5. Can the authors characterize the filtration of neutrophil cells to understand the targeted therapy of acute lung inflammation?
6. The author mentioned that the synergistic effect of drugs (DS or QT) and GMs, and the characterization data such as synergy coefficient are missing.
7. Please provide the scale bars of Fig. S6a and Fig. S8a.
8. The caption in Figure 7 is inconsistent with the presented content.

Reviewer #2 (Remarks to the Author):

Gao et al reported a study of "Intracellularly gelated macrophages for targeted therapy of inflammatory diseases. It is an interesting, but there are some technical concerns and novelty issues. Please see the comments below.

- 1) The studies do not show the advantages of using gelated macrophages over membrane-derived nanoscale vesicles or NPs without the comparison data. Fig. 3F shows that the neutralization of cytokines using GMs was likely not effective. It is not clear that GMs are better than membrane-derived vesicles.
- 2) The authors proposed to gelate macrophages to maintain cellular membrane proteins and load drugs inside gelated macrophages. These macrophages are microsized, and they accumulate in the liver (such as Fig. 7A and B). It looks like that they had a short circulation

time, so it had the less efficacy to decoy cytokines in blood. In addition, gelated macrophages are large and rigid. How gelated macrophages cross lung vasculature to deliver drugs inside airspace of the lung?

3) The authors missed the control of non-targeted cells, such as red blood cells because they do not have the cytokine decoys. This is important for lung targeting studies.

4) The studies in Fig. 5 missed the controls of DS+anti-TNF α .

Minors: Reference 19 is not relevant to the current studies.

Reviewer #3 (Remarks to the Author):

In the manuscript "Intracellularly gelated macrophages for targeted therapy of inflammatory diseases" submitted by Cheng Gao et al. the authors propose a method for generating chelated macrophages that allows for cytokine scavenging and drug delivery in inflamed tissue. In the paper, the authors initially present some basic characteristics of the chelated macrophages, showing among others that the chelation process leads to cell death while retaining the size and the architecture of the macrophages and may sequester inflammatory cytokines in vitro.

Subsequently the chelated macrophages are loaded with diclofenac sodium or Quercetin and tested in vivo in two different inflammatory models. Although the paper is of some interest, a very large amount of basic information and characterization is missing before the manuscript should be considered for publication.

Some comments.

1. What is the source of macrophages used in this study? In the materials section RAW264.7 cells are mentioned. However, since this is a mouse macrophage-type cell line, it is of very low therapeutic relevance. Here bone-marrow derived macrophages or monocyte derived macrophage would have been of much higher relevance. Also, as the macrophage phenotype is fixed after chelating, it would have been relevant to test the effect of different macrophage phenotypes on cytokine level.

2. In figure 3 the dose of chelated macrophages is given in mg/ml, however in the CIA treatment study the dose is mentioned as 3×10^6 cells per rat. In order to compare the

results and assess the relevance of the chelated macrophages as a therapeutic drug all doses should be given in number of cells.

3. There is no characterization on the drug loading in chelated macrophages and in addition the difference in treatment effect in figure 5 between GMs and DS-GMs is so small that it is hard to judge if there is a treatment effect. How does the GMs compare to glucocorticoids that are normally also administered locally?

4. What is the source of the anti-TNF- α used in figure 5? In the text it is stated that anti-TNF- α is included as positive control, however with the dose given in this study there seems to be no effect. Was the anti-TNF- α also administered in the footpad? In mice a normal dose of 10 mg/kg anti-TNF- α administered i.p or i.v. often have a strong effect on disease progression (e.g. Shealy DJ et al. *Arthritis Res.* 2002;4(5):R7)

5. In the lung inflammation model the effect is measured 6h after injection of GMs. However, given the assumed slow release of drugs from GMs this time point seems extremely early. Also acute infiltration of monocytes are believed to peak after ca. 24h and 6h would therefore be too early to study the effect of the GM on recruitment of inflammatory macrophages (Kratofil RM et al. *Arterioscler Thromb Vasc Biol.* 2017 Jan;37(1):35-42.). Also there is no mention of a live/dead marker to exclude dead cells in the flow analysis performed in relation to this experiment. Consequently, no conclusion can be drawn from this experiment

Response Letter

Reviewer #1 (Remarks to the Author):

In this paper, the authors developed an intracellularly gelated macrophages (GMs) with highly preserved cell membrane structures, which can function as a natural antiinflammation agent and stable drug carrier to treat inflammatory diseases of rheumatoid arthritis and acute pneumonia. Although an interesting work, there are a few scientific shortcomings. It needs to carry out some additional experiments to confirm some of the results in this work.

Response: Thanks for the valuable comments. We have conducted additional experiments to provide more solid evidence to confirm our claims in this research, which have significantly helped improve the quality of this research.

1. Please highlight the novelty by comparing with previous works.

Response: Thanks for the suggestion. Accordingly, the novelty of this work has been described in the conclusion section by comparing with previous works (**Page 21**).

“Due to the receptor-ligand interaction, cell membrane camouflaged nanomedicine has developed rapidly in recent years, and they could also absorb and neutralize multiple pathological molecules.^{46,47} However, membrane protein disintegration, loss and spatial disorder are inevitable problems during the construction of membrane coated nanomedicine, which reduces their neutralization efficiency. Thus, one of the highlights of this work is that the gelated cell highly preserved intact cell membrane structures, membrane protein mobility and membrane lipid order. Gelated cell was more effective to neutralized cytokines and chemokines in comparison to membrane coated nanomedicine under the same number of source cells. Secondly, intracellular hydrogel could be utilized for drug loading, and gelated cell could act as cell-based drug carrier for hitchhiking delivery. This design has addressed the main challenges of drug metabolism and efflux in the current living cell-based carriers. Finally, clinical monoclonal antibody only inhibits one or a few of pathological molecules, and most of them are stored at low temperatures. Gelated cell inherits intact protein components of source cell and shows a high neutralization efficiency towards multiple pathological molecules. Hydrogel cytoskeleton has endowed gelated cell with capability for freeze drying treatment that allows storage and transportation at a room temperature, which will benefit potential clinical translation.”

2. For in vivo experiments, the MAs and hydrogel are better to be used as control.

Response: Thanks for the insightful advice. Accordingly, we have supplemented MAs and hydrogel as control groups in the treatment of arthritis and acute pneumonia. As shown in **Supplementary Fig. 5 and Fig. 6**, the therapeutic efficacies of both MAs and

hydrogel on CIA rat were similar to that of saline, and no obvious improvements in the symptom of joint injury were observed (**Page 13**). In the treatment of AP mice, both MAs and hydrogel showed negligible anti-inflammation effect, and MAs treatment even slightly aggravated lung tissue inflammation (**Supplementary Fig. 11**) (**Page 20**). These results further confirmed that GMs alleviated arthritis through the cytokine sequestration via membrane rather than the intracellular hydrogel, and living MAs could be activated to increase inflammation, whereas GMs absorbed cytokines without any risk of activation.

Supplementary Fig. 5. Photographs of the rat paws taken at all time points in different treatment groups during therapeutic treatment. The swelling paws of CIA rats after 4 weeks of arthritis development were injected with saline, DS, GMs, DS-GMs, aTNF- α , and DS+ aTNF- α at 3×10^6 of GMs per rat, 1 mg kg^{-1} of DS and 1 mg kg^{-1} of aTNF- α . Rats treated with 2 mg kg^{-1} of hydrogel, 3×10^6 of MAs and 2 mg kg^{-1} of PSL served control groups. After treatment once every four days until Day 60, representative photos at determined time points demonstrated the progression of mouse paw inflammation.

Supplementary Fig. 6. Therapeutic efficacy of hydrogel, MAs, DS+aTNF- α and GC in CIA rats. (A) The changes of paw thickness in CIA rats after treatments with DS+ aTNF- α at 1 mg kg⁻¹ of DS and 1 mg kg⁻¹ of aTNF- α , 2 mg kg⁻¹ of hydrogel, 3x10⁶ of MAs and 2 mg kg⁻¹ of PSL. (B and C) Representative images of Fibronectin staining (B) and CD248 staining (C) on the on the synovial intimal lining from treated rats. Scale bar: 100 μ m. (D) The concentrations of serum TNF- α and IL-1 β at the endpoint of treatment. (E) The ankle diameter changes of treated rats compared to that of the normal rats recorded at Day 52. (F) The overall arthritis score of the treated rats recorded every four days with a total of 28 days after arthritis induction. All data was presented as mean \pm s.d. (n = 3). All statistical analyses were conducted by using One-Way ANOVA. *P \leq 0.05 and ***P \leq 0.001.

Supplementary Fig. 6. Therapeutic efficacy of hydrogel, MAs, DS+aTNF- α and GC in CIA rats. (A) The changes of paw thickness in CIA rats after treatments with DS+ aTNF- α at 1 mg kg⁻¹ of DS and 1 mg kg⁻¹ of aTNF- α , 2 mg kg⁻¹ of hydrogel, 3x10⁶ of MAs and 2 mg kg⁻¹ of PSL. (B and C) Representative images of Fibronectin staining (B) and CD248 staining (C) on the on the synovial intimal lining from treated rats. Scale bar: 100 μ m. (D) The concentrations of serum TNF- α and IL-1 β at the endpoint of treatment. (E) The ankle diameter changes of treated rats compared to that of the normal rats recorded at Day 52. (F) The overall arthritis score of the treated rats recorded every four days with a total of 28 days after arthritis induction. All data was presented as mean \pm s.d. (n = 3). All statistical analyses were conducted by using One-Way ANOVA. *P \leq 0.05 and ***P \leq 0.001.

3. The DS, QT and hydrogel loading contents of GMs should be provided.

Response: Thanks. Accordingly, the drug loading capacity of DS-GMs used for anti-arthritis treatment was determined to be ~0.4%, suggesting that ~0.004 mg DS loaded per mg of DS-GMs. In the treatment of acute pneumonia, the weight content of QT in

QT-GMs was ~1.2%, indicating that the loading capacity of QT-GMs was ~1.2%. This information has been supplemented in the revised manuscript (**Page 23**). Because it was hard to separate the intracellular hydrogel from cell component, thus the percent of hydrogel weight in GMs could not be measured.

4. The swelling properties of hydrogels require further evaluation.

Response: Thanks for your suggestion, and supramolecular hydrogel was prepared from 2% (wt%) of Phe-CS and 50 μ M of CB[8] for the measurement of swelling ratio, which was the same concentration utilized to prepare GMs. As shown in **Supplementary Fig. 1C**, the hydrogel swelled quickly at the beginning after placement in PBS solution, indicating a fast drug release behavior when utilized for drug loading (**Fig. 2G**). Then, it reached equilibrium on Day 5 with a maximum swelling ratio of 150%.

Supplementary Fig. 1C. Swelling ratio of hydrogel after placement in PBS for different times (1 day, 2, 3, 4, 5, 6, and 7 days).

5. Can the authors characterize the filtration of neutrophil cells to understand the targeted therapy of acute lung inflammation?

Response: Thanks for the insightful suggestions. Accordingly, immunofluorescent staining of neutrophil cells (CD11b⁺Gr-1⁺ cells) was conducted in lung tissues of treated AP mice. As shown in **Supplementary Fig. 12**, the fluorescent intensity of CD11b⁺Gr-1⁺ cell was strong in AP mice treated with MAs (~16.2%) and hydrogel (~15%), which was comparable to saline treated mice (~15.8%). GMs and QT both decreased the fluorescent neutrophils in lung sections, and a significant reduction of neutrophil infiltration was detected in QT-GMs treated mice (~4.7%), attributed to the combined anti-inflammation effect of QT and GMs (**Page 20**).

Supplementary Fig. 12. Fluorescence imaging on the filtration of neutrophils (CD11b⁺Gr-1⁺ cells) in the lungs of treated mice, and semi-quantitative analysis by ImageJ software. Scale bar: 40 μ m.

6. The author mentioned that the synergistic effect of drugs (DS or QT) and GMs, and the characterization data such as synergy coefficient are missing.

Response: Sorry for the inaccurate description. The effects of drugs (DS or QT) and GMs have been revised into the “combined therapy of drugs (DS or QT) and GMs” throughout the manuscript. Anti-inflammatory drugs (DS or QT) in this study were selected for assisting the cytokine neutralization effect of GMs in the treatment of arthritis and acute pneumonia, as well as to prove the advantage of GMs served as “cell-based carrier”. DS or QT inhibited inflammation via regulating inflammatory signaling pathway, while GMs directly absorbed proinflammatory cytokines and chemokines to reduce inflammation. This work simply compared the therapeutic efficacy of drugs (DS or QT), GMs and drug loaded GMs at the same dose of drugs and GMs, and did not evaluate the different therapeutic outcomes after treatment with different doses of drugs (DS or QT), GMs and drug loaded GMs. Thus, the conclusion of synergistic effect of drugs and GMs could not be drawn. Description of the synergistic effect of drugs (DS or QT) and GMs has been revised into the “combined therapy of drugs (DS or QT) and GMs” throughout the manuscript.

7. Please provide the scale bars of Fig. S6a and Fig. S8a.

Response: Thanks for the kind reminder, and the scale bars in Supplementary Fig. 6a (updated **Supplementary Fig. 8A**) and Supplementary Fig. 8A (updated **Supplementary Fig. 13a**) have been supplemented.

8. The caption in Figure 7 is inconsistent with the presented content.

Response: Sorry for our negligence. **Fig. 7H** and **7I**, and their corresponding captions were opposite to the presented content, and we have corrected their order in the revised **Fig. 7** and corresponding caption.

Reviewer #2 (Remarks to the Author):

Gao et al reported a study of “Intracellularly gelated macrophages for targeted therapy of inflammatory diseases. It is an interesting, but there are some technical concerns and novelty issues. Please see the comments below.

Response: Thanks a lot for your careful review and valuable comments. The main technical concerns and novelty issues of the work have been addressed by explicit explanation or supplemented experiments in the following points of responses to your comments.

1) The studies do not show the advantages of using gelated macrophages over membrane-derived nanoscale vesicles or NPs without the comparison data. Fig. 3F shows that the neutralization of cytokines using GMs was likely not effective. It is not clear that GMs are better than membrane-derived vesicles.

Response: Thanks for the suggestion, and we supplemented the experiments to compare the cytokine neutralization efficiency of GMs and macrophage membrane coated nanoparticles (MM-NPs). Macrophage membrane was collected from macrophage and then coated on PLGA NPs through extrusion method to prepare MM-NPs. GMs and MM-NPs prepared from same number of macrophages (3×10^6) were incubated in 1 mL of PBS solution containing 10 ng mL^{-1} of TNF- α and 10 ng mL^{-1} of IL-1 β for 1 h, respectively. After removing the precipitates, the supernatant cytokine concentrations were measured by Elisa kits. As shown in **Supplementary Fig. 4A** and **4B**, GMs and MM-NPs significantly reduced the TNF- α level into 2.93 and 6.87 $\mu\text{g mL}^{-1}$, and IL-1 β level into 2.36 and 6.14 $\mu\text{g mL}^{-1}$ in supernatant, but the remaining TNF- α and IL-1 β concentration in MM-NPs treated group was ~ 2.45 -fold and ~ 2.6 -fold higher than that of GMs treated group, respectively. These results exhibited a strong cytokine sequestration ability of GMs in comparison to MM-NPs. In addition, the total membrane protein content in GMs and MM-NPs prepared from same number of macrophages (3×10^6) were detected by BCA protein assay. **Supplementary Fig. 4C** exhibited the total membrane protein content of GMs was $3 \times 10^{-3} \mu\text{g}$, which was ~ 1.86 -fold than that of MM-NPs ($1.61 \times 10^{-3} \mu\text{g}$), indicating that the membrane protein loss led to a low cytokine sequestration efficiency. Regarding the 1.86-fold of membrane protein loss and 2.45-fold or 2.6-fold of decreased neutralization efficiency of MM-NP, the protein spatial disorder during membrane re-assembly process might reduce the cytokine neutralization ability of MM-NP. Collectively, these evidences showed the

intact membrane structure of GMs contributed to a better cytokine neutralization efficiency (Page 9).

Supplementary Fig. 4. Cytokine neutralization efficiency of GMs and MM-NPs. (A and B) GMs and MM-NPs prepared from same number of macrophages (3×10^6) were incubated in 1 mL of PBS solution containing 10 ng mL^{-1} of TNF- α (A) and 10 ng mL^{-1} of IL-1 β (B) for 1 h, respectively. After removing the precipitates, the supernatant cytokine concentrations were measured by Elisa kits. (C) Membrane protein contents of GMs and MM-NPs prepared from same number of macrophages (3×10^6) were analyzed by BCA protein assay. All data was presented as mean \pm s.d. ($n = 3$). All statistical analyses were conducted using One-Way ANOVA. * $P \leq 0.05$ and *** $P \leq 0.001$.

2) The authors proposed to gelate macrophages to maintain cellular membrane proteins and load drugs inside gelated macrophages. These macrophages are micro-sized, and they accumulate in the liver (such as Fig. 7A and B). It looks like that they had a short circulation time, so it had the less efficacy to decoy cytokines in blood. In addition, gelated macrophages are large and rigid. How gelated macrophages cross lung vasculature to deliver drugs inside airspace of the lung?

Response: Thanks for the insightful comments. Accordingly, we have supplemented new experiments to study the blood circulation of GMs. Macrophages were stained by DiD (membrane red fluorescent dye) before intracellular gelation (DiD-GMs) to allow tracking of their in vivo biodistribution. After *i.v.* administration with 3×10^6 of DiD-GMs for different durations (1, 2, 3, 4, 5 and 6 h), the peripheral circulation blood was collected from AP mice for ex vivo fluorescence imaging. As shown in **Supplementary Fig. 9C**, to our surprises, negligible fluorescence was detected even after administration for 1 h, suggesting that GMs had a very short circulation time. Based on the results of **Fig. 7A** and **7B**, GMs possessed macrophage-like inflammatory tropism to the lung tissue. Considering that GMs is rigid and does not have the plasticity of living macrophages, it is indeed hard to penetrate through lung vasculature, and most of them are likely located inside the abundant lung capillaries. Thus, GMs neutralizes cytokines in the lung capillary, released from inflammatory pneumonocyte. In addition to lung tissue, GMs were mainly distributed in liver tissue. When blood flew through these tissues, GMs also sequestered the cytokines in blood and exhibited the systemic anti-

inflammation effect. Moreover, the accumulated drug release from GMs reached ca. 65% after incubation in PBS for 6 h (**Fig. 2G**), indicating a quick drug release behavior. After accumulation in inflammatory lung tissue, the shear stress in abundant lung capillaries could push out the intracellular drug from GMs more quickly, and then these drugs went through lung capillaries and penetrated into inflammatory pneumonocyte for anti-inflammation therapy. This discussion was added in the manuscript (**Page 19** and **Page 20**).

Supplementary Fig. 9C. AP mice were *i.v.* administered with 3×10^6 of DiD-GMs for different times (1 h, 2 h, 3 h, 4 h, 5 h and 6 h), and peripheral blood was collected for ex vivo fluorescence imaging.

3) The authors missed the control of non-targeted cells, such as red blood cells because they do not have the cytokine decoys. This is important for lung targeting studies.

Response: Thanks for the advice. Accordingly, in the lung targeting study of GMs in AP mice, red blood cells (RBCs) were supplemented as non-targeted cell model. AP mice were *i.v.* administered with 0.5 mg kg^{-1} of Cy5 loaded RBCs (Cy5-RBCs), and lung tissues were collected after injection for 10 min for ex vivo fluorescent imaging (**Supplementary Fig. 9**). A much weaker fluorescence intensity was observed in Cy5-RBCs treated mice in comparison to that of Cy5-GMs treated mice, suggesting RBCs has negligible inflammatory tropism. Nevertheless, the shear stress in abundant lung capillaries could help push out the intracellular Cy5 from RBCs, contributing to a slightly brighter fluorescence intensity in comparison to free Cy5 (**Page 18**).

Supplementary Fig. 9. Ex vivo biodistribution of Cy5-RBCs in AP mice. (A and B) AP mice were *i.v.* administered with 0.5 mg kg^{-1} of Cy5. After injection for 10 min, different organs (heart, liver, spleen, lungs and kidneys) were collected for

ex vivo fluorescence imaging (A), and the fluorescence intensity was quantified by IVIS (B).

4) The studies in Fig. 5 missed the controls of DS+anti-TNF α .

Response: Thanks for the suggestion, and CIA rat treated with 1 mg kg⁻¹ of DS and 1 mg kg⁻¹ of aTNF- α (DS+aTNF- α) was supplemented as control group. As shown in **Supplementary Fig. 5** and **6**, the combination of DS and aTNF- α exhibited a better anti-arthritis efficacy than that of DS treatment and aTNF- α treatment. Although the therapeutic efficacy of aTNF- α was comparable to that of GMs, but DS+aTNF- α was still slightly weaker in comparison to DS-GMs, attributed to that GMs increased the retention time of loaded DS in the swelling paw (**Fig. 5A** and **5B**). This indicated that GMs not only had the cytokine sequestration effect, but also served as cell-carrier to increase the accumulation of loaded drug in the swelling paw for a combined therapy (**Page 13**).

Supplementary Fig. 5. Photographs of the rat paws taken at all time points in different treatment groups during therapeutic treatment. The swelling paws of CIA rats after 4 weeks of arthritis development were injected with saline, DS, GMs, DS-GMs, aTNF- α , and DS+ aTNF- α at 3×10^6 of GMs per rat, 1 mg kg⁻¹ of DS and 1 mg kg⁻¹ of aTNF- α . Rats treated with 2 mg kg⁻¹ of hydrogel, 3×10^6 of MAs and 2 mg kg⁻¹

of PSL served control groups. After treatment once every four days until Day 60, representative photos at determined time points demonstrated the progression of mouse paw inflammation.

Supplementary Fig. 6. Therapeutic efficacy of hydrogel, MAs, DS+aTNF- α and GC in CIA rats. (A) The changes of paw thickness in CIA rats after treatments with DS+ aTNF- α at 1 mg kg⁻¹ of DS and 1 mg kg⁻¹ of aTNF- α , 2 mg kg⁻¹ of hydrogel, 3x10⁶ of MAs and 2 mg kg⁻¹ of PSL. (B and C) Representative images of Fibronectin staining (B) and CD248 staining (C) on the on the synovial intimal lining from treated rats. Scale bar: 100 μ m. (D) The concentrations of serum TNF- α and IL-1 β at the endpoint of treatment. (E) The ankle diameter changes of treated rats compared to that of the normal rats recorded at Day 52. (F) The overall arthritis score of the treated rats recorded every four days with a total of 28 days after arthritis induction. All data was presented as mean \pm s.d. (n = 3). All statistical analyses were conducted by using One-Way ANOVA. *P \leq 0.05 and ***P \leq 0.001.

Minors: Reference 19 is not relevant to the current studies.

Response: Sorry for the negligence. Reference 19 has been changed into a relevant

literature (Advanced Materials, 2018, 30, 1805557), which title was “Nanoparticle-laden macrophages for tumor-tropic drug delivery”.

Reviewer #3 (Remarks to the Author):

In the manuscript “Intracellularly gelled macrophages for targeted therapy of inflammatory diseases” submitted by Cheng Gao et al. the authors propose a method for generating chelated macrophages that allows for cytokine scavenging and drug delivery in inflamed tissue. In the paper, the authors initially present some basic characteristics of the chelated macrophages, showing among others that the chelation process leads to cell death while retaining the size and the architecture of the macrophages and may sequester inflammatory cytokines in vitro.

Subsequently the chelated macrophages are loaded with diclofenac sodium or Quercetin and tested in vivo in two different inflammatory models. Although the paper is of some interest, a very large amount of basic information and characterization is missing before the manuscript should be considered for publication.

Response: Thanks a lot for your kind review and valuable comments. The missing information and characterization proposed by the reviewer have been addressed with clear description and supplemented experiments, which have helped to significantly improve the integrity and quality of this research.

Some comments.

1. What is the source of macrophages used in this study? In the materials section RAW264.7 cells are mentioned. However, since this is a mouse macrophage-type cell line, it is of very low therapeutic relevance. Here bone-marrow derived macrophages or monocyte derived macrophage would have been of much higher relevance. Also, as the macrophage phenotype is fixed after chelating, it would have been relevant to test the effect of different macrophage phenotypes on cytokine level.

Response: Thanks for the suggestion, and we agreed with reviewer’s opinion that a much higher relevance macrophage, such as bone-marrow derived macrophage or monocyte derived macrophage, might contribute to a better therapeutic efficacy. In this study, membrane protein analysis on GMs prepared from RAW264.7 cells showed the abundance of key inflammatory receptors, including tumor necrosis factor receptors (TNFR1 and TNFR2), interleukin 6 receptors (IL6R and IL6RB), and interleukin 1 receptor (IL1R2) (**Fig. 3C** and **3D**). Thus, these GMs were utilized for cytokine sequestration and in vivo anti-inflammation study. The results of cytokine neutralization curve, arthritis prevention and therapy, and lung inflammation alleviation really did support the significant cytokine neutralization and anti-inflammation efficiency of these GMs. The data indeed speak for themselves. Thus, the anti-inflammation effect of GMs might be mainly dependent on the membrane inflammatory receptors. Because the bone-marrow derived macrophage and monocyte derived

macrophage was more prone to body cytokines and chemokines, GMs prepared from those cells might contribute to a more appropriate affinity to multiple pathogens. We have added this discussion in the conclusion section of manuscript (**page 21**).

The macrophage type used in this study was M0. As the reviewer's comment, macrophage phenotypes really affected the cytokine sequestration efficiency of GMs. Recently, we are conducting a new project, and find that macrophage educated by LPS or bacteria before intracellular gelation increases the sequestration efficiency towards proinflammatory cytokines (TNF- α and IL-1 β) or bacteria (E. coli and S. aureus) in comparison to GMs prepared from M0 macrophages. In contrast, macrophage pretreated with IL-4 before intracellular gelation, the cytokine or bacteria binding capability was even high than that of GMs prepared from M1 macrophages. This should be attributed to that M1 macrophage increased the production of cytokine-related receptors or Toll-like receptors after cytokine or bacteria education, and M2 macrophage produced more cytokine-related receptors. Because this is our on-going project, the relevant data will be published in future with more solid data collected. For the past over 5-month, we have spent all the efforts to supplement data to satisfy all of the reviewers' comments.

2. In figure 3 the dose of chelated macrophages is given in mg/ml, however in the CIA treatment study the dose is mentioned as 3×10^6 cells per rat. In order to compare the results and assess the relevance of the chelated macrophages as a therapeutic drug all doses should be given in number of cells.

Response: Thanks for the suggestion, and we have changed all the description (including **Fig. 3**) on the dose of GMs into the number of cells, and made it comparable and consistent throughout the manuscript.

Fig. 3. Membrane protein integrity and fluidity of GMs enhanced cytokine sequestration efficiency. (E) Different doses of GMs (0.15, 0.3, 0.6, 1.2, 2.4, 4.8 and 9.6, 10⁶ mL⁻¹) were incubated with PBS solution containing 10 μ g mL⁻¹ each of TNF- α and IL-1 β , and the supernatant concentrations of TNF- α and IL-1 β were measured by Elisa kits after incubation for 1 h. (F) MAs were co-treated with 100 ng mL⁻¹ of LPS and different dose of GMs (0.1875, 0.375, 0.75, 1.5 and 3, 10⁶ mL⁻¹) for 12 h, and the supernatant concentrations of TNF- α and IL-1 β were measured by Elisa kits. (G) Schematic illustration of IL-6 trans-signaling pathway. (H) MAs, GMs and PFA-MAs were incubated with 10 ng mL⁻¹ of IL-6 for 12 h, and incubated with rabbit gp130 primary antibody and mouse IL-6R primary antibody, respectively, followed by incubation with anti-rabbit IgG Fab2 Alexa Fluor 488 and anti-mouse IgG Fab2 Alexa Fluor 594, fluorescence imaging was conducted by confocal microscopy. Scale bar: 5 μ m. (I) Pearson's correlation coefficient of fluorescence co-localization was determined by ImageJ. (J) MAs, GMs and PFA-MAs at the number of 3x10⁶ were incubated with 10 μ g mL⁻¹ of IL-6 for 12 h, and the remaining IL-6 concentration was determined by Elisa kit. All data was presented as mean \pm s.d. (n = 3). All statistical analyses were conducted using One-Way ANOVA. **P \leq 0.01 and ***P \leq 0.001.

3. There is no characterization on the drug loading in chelated macrophages and in addition the difference in treatment effect in figure 5 between GMs and DS-GMs is so small that it is hard to judge if there is a treatment effect. How does the GMs compare to glucocorticoids that are normally also administrated locally?

Response: The drug loading capacity of DS-GMs used for anti-arthritis treatment was ~0.4%, indicating that ~0.004 mg DS loaded per mg of DS-GMs. In the treatment of acute pneumonia, the weight content of QT in QT-GMs was ~1.2%, indicating that the loading capacity of QT-GMs was ~1.2%. This information has been supplemented in the manuscript (**Page 23**).

Per the reviewer's comment, in **Fig. 5** GD-GMs showed a litter better performance than GMs. However, both GMs and DS-GMs efficiently reduced paw redness and thickness, cartilage content, serum inflammation level and the ankle diameter change, and GMs treated rats also had a low overall arthritis score in comparison to that of aTNF- α and saline treated groups (**Fig. 5J**). These results really did support that GMs efficiently inhibited the pathological progression of arthritis, attributed to that the intact membrane structures, including multiple cytokine receptors, protein fluidity and lipid order. DS inhibited inflammation via regulating inflammatory signaling pathway, while GMs directly absorbed proinflammatory cytokines to reduce inflammation. After combination with anti-inflammatory DS, the therapeutic efficacy of GMs was not significant improved, but DS-GMs treated rats still exhibited the lowest overall arthritis score. This result indicated that the final treatment effect of DS-GMs was not dependent on the simple addition of cytokine neutralization of GMs and anti-inflammation of DS.

Finally, to compare the anti-arthritis efficacy of locally administered glucocorticoids and GMs, CIA rats treated with 2 mg kg⁻¹ of prednisolone (an anti-inflammatory glucocorticoid) served as positive control group. As shown in **Supplementary Fig. 5** and **6**, prednisolone (PSL) suppressed arthritis significantly, compared to GMs, and similar to that of DS-GMs. The result showed the powerful anti-arthritis effect of PSL. However, GMs exhibited ant-arthritis effect through multiple cytokines neutralization, unlike PSL via suppression of the migration of polymorphonuclear leukocytes and reversing increased capillary permeability. Glucocorticoid-induced severe adverse effects such as osteoporosis, metabolic disturbances (diabetes, atypical fat accrual, and dyslipidemia), skin atrophy, and hypertension has led to recommendations that glucocorticoids only be used at low doses over short periods of time (Nature Reviews Rheumatology, 2020, 16(4), 239-246). Thus, GMs prepared from self-derived macrophages exhibited higher biosafety than glucocorticoids and a better therapeutic efficacy than clinical monoclonal antibody drugs (such as anti-TNF- α , anti-IL-1 β), making them an ideal substitution for long time treatment of arthritis (**Page 13**).

Supplementary Fig. 5. Photographs of the rat paws taken at all time points in different treatment groups during therapeutic treatment. The swelling paws of CIA rats after 4 weeks of arthritis development were injected with saline, DS, GMs, DS-GMs, aTNF- α , and DS+ aTNF- α at 3×10^6 of GMs per rat, 1 mg kg^{-1} of DS and 1 mg kg^{-1} of aTNF- α . Rats treated with 2 mg kg^{-1} of hydrogel, 3×10^6 of MAs and 2 mg kg^{-1} of PSL served control groups. After treatment once every four days until Day 60, representative photos at determined time points demonstrated the progression of mouse paw inflammation.

Supplementary Fig. 6. Therapeutic efficacy of hydrogel, MAs, DS+aTNF- α and GC in CIA rats. (A) The changes of paw thickness in CIA rats after treatments with DS+ aTNF- α at 1 mg kg⁻¹ of DS and 1 mg kg⁻¹ of aTNF- α , 2 mg kg⁻¹ of hydrogel, 3x10⁶ of MAs and 2 mg kg⁻¹ of PSL. (B and C) Representative images of Fibronectin staining (B) and CD248 staining (C) on the on the synovial intimal lining from treated rats. Scale bar: 100 μ m. (D) The concentrations of serum TNF- α and IL-1 β at the endpoint of treatment. (E) The ankle diameter changes of treated rats compared to that of the normal rats recorded at Day 52. (F) The overall arthritis score of the treated rats recorded every four days with a total of 28 days after arthritis induction. All data was presented as mean \pm s.d. (n = 3). All statistical analyses were conducted by using One-Way ANOVA. *P \leq 0.05 and ***P \leq 0.001.

4. What is the source of the anti-TNF- α used in figure 5? In the text it is stated that anti-TNF- α is included as positive control, however with the dose given in this study there seems to be no effect. Was the anti-TNF- α also administrated in the footpad? In mice a normal dose of 10 mg/kg anti-TNF- α administrated i.p. or i.v. often have a strong effect on disease progression (e.g. Shealy DJ et al. Arthritis Res. 2002;4(5):R7)

Response: Thanks for the comments. aTNF- α used in this study was purified anti-mouse/rat TNF- α antibody purchased from Biolegend (Catalog number: 506114), and it was administered in the footpad of CIA rat. This information was added in the manuscript (**Page 12**). In **Fig. 5**, statistical analysis was conducted between PBS group and anti-TNF- α group, and the results showed there was a statistical difference. In comparison to PBS, anti-TNF- α significantly decreased paw redness and thickness, low systemic inflammation and reversal of cartilage degradation. In the literature (Shealy DJ et al. Arthritis Res. 2002, 4(5), R7), after anti-TNF- α treatment (10 mg kg⁻¹) for 6 times, improvements in all pathological indicators were observed, and no visible evidence of bone or cartilage erosion was found. The proteoglycan content of the cartilage was notably improved. In addition, the literature also showed that extended treatment with anti-TNF- α for 16 times reversed all pathological indicators to normal level. Actually, rats were treated with 10 mg kg⁻¹ of anti-TNF- α every four days with a total of 7 times in our study. It was not as effective as the mice treated with anti-TNF- α for 16 times in the literature, but the therapeutic efficacy in our study was consistent with the reported results of anti-TNF- α treatment for 6 times, both of which reduced the overall arthritis score from over 4 to about 2. The very modest differences should be attributed to the different species of SD rats in this study and heterozygous Tg197 transgenic mice in the reported literature.

5. In the lung inflammation model the effect is measured 6h after injection of GMs. However, given the assumed slow release of drugs from GMs this time point seems extremely early. Also acute infiltration of monocytes are believed to peak after ca. 24h and 6h would therefor be to early to study the effect of the GM on recruitment of inflammatory macrophages (Kratofil RM Et al. Arterioscler Thromb Vasc Biol. 2017 Jan;37(1):35-42.). Also there is no mention of a live/dead marker to exclude dead cells in the flow analysis performed in relation to this experiment. Consequently, no conclusion can be drawn from this experiment.

Response: Thanks for the comments, and accordingly we supplemented the explanations and experiments to address the concerns about administration time and flow cytometry analysis in the AP model. Drug release profile (**Fig. 2G**) showed that the accumulative release ratio of drug loaded GMs reached ~60% after incubation in PBS for 6 h, indicating most drug was released for anti-AP treatment with QT-GMs. For the administration time in the treatment of acute pneumonia (AP) mouse, we agreed with the reviewer's point that acute infiltration of monocytes was believed to peak after ca. 24 h (Kratofil RM Et al. Arterioscler Thromb Vasc Biol. 2017, 37(1), 35-42), and thereby treatment with GMs for 24 h would contribute to a better therapeutic efficacy on AP mouse in comparison to that of 6 h. Nevertheless, the purpose of this anti-acute pneumonia experiment was to evaluate the GM-hitchhiking delivery and the combined anti-inflammatory effect of GMs and QT. In vivo biodistribution study showed a high accumulation rate of GMs in inflammatory lung tissue only after *i.v.* administration for 10 min (**Fig. 5A** and **5B**), indicating a quick response to inflammatory lung. GMs obviously alleviated lung inflammation symptom at 6 h (**Fig. 5C-J**), and QT-GMs

exhibited best performance due to GM-hitchhiking delivery to inflammatory lung and combined anti-inflammatory therapy of GMs and QT. Our results of different therapeutic efficacies between saline, GMs, QT, and QT-GMs really did support the claims of “cell sponge” and “immune cell-like carrier” of GMs for efficiently targeted treatment on AP mice, although the inflammation symptom was not completely cured and treatment for 24 h might contribute to a better therapeutic outcome on AP mouse.

Per the reviewer’s suggestion of “there was no mention of a live/dead marker to exclude dead cells in the flow analysis”, we re-conducted this experiment and stained the collected cells with LIVE/DEAD™ Fixable Near-IR Dead Cell Stain Kit to exclude dead cells. As shown in **Supplementary Fig. 10** (see next page), LPS induced the significant filtration of F4/80⁺ cells (MAs) and locally polarized F4/80⁺CD86⁺ cells (M1 MAs) in the inflammatory lung tissues (**Fig. 7K** and **7L**). The ratio of MA infiltration was reduced in mice treated with GMs, and MA polarization into M1 type was also inhibited, comparable to that of QT treated mice. QT-GMs exhibited the most pronounced therapeutic outcome in the MA infiltration and polarization, when compared with other therapeutic groups, confirming that QT-GMs may effectively ameliorate the acute lung inflammation via both cytokine neutralization of GMs and anti-inflammatory effects of QT.

Supplementary Fig. 10. Gating strategy of flow cytometry analysis on the filtration of macrophages ($F4/80^+CD11b^+$ cells) and M1 macrophages ($F4/80^+CD11b^+CD86^+$ cells) in lung tissues. AP mice was blindly and randomly divided into four groups ($n=3$), and *i.v.* administered with Saline, QT, QT-GMs, and GMs at 5×10^6 of GMs per mice, 5 mg kg^{-1} of QT, 2 mg Kg^{-1} of hydrogel and 3×10^6 of MAs. After administration for 6 h, the lung tissues were collected for flow cytometry analysis.

REVIEWER COMMENTS

Reviewer #1 (Remarks to the Author):

Although the resubmitted manuscript adds a lot of data, there are still some questions that have not been seriously answered.

1. The swelling properties of hydrogels require further evaluation.

Authors only examined the swelling ratio of hydrogels, which does not account for swelling properties. Confusingly, the authors concluded that the maximum swelling ratio of the gel was achieved after 5 days, while the GM remained intact in the body for up to 5 days, which seems difficult to achieve.

2. Authors suggested that the drug loading capacity of DS-GMs used for anti-arthritis treatment was ~0.4%. In the treatment of acute pneumonia, the weight content of QT in QT-GMs was ~1.2%. It is amazing that such a low drug load can achieve such good results, so the relevant mechanism research must be performed. In addition, is my concern that such a good treatment effect has similar effects in other inflammatory models?

3. In vivo pharmacokinetic studies need to be provided.

4. Notably, the assessment of macrophages is overly simplistic given the highly controversial nature of M1 and M2 classification (as well as the M1/M2 binary relevance to in vivo function, see <https://doi.org/10.1186/s12885-018-4457-8> and <https://doi.org/10.12703/P6-13> and <https://doi.org/10.1038/nri3088> and <https://doi.org/10.1016/j.addr.2017.05.010>). Further, with regards to the effect on the macrophages population, it is unclear if the effects are due to polarization of the macrophage cells or deletion of immunosuppressive phenotypes.

Reviewer #2 (Remarks to the Author):

The authors provided some new data to address the reviewer's questions, but there are still some concerns.

1. The authors compared the lung accumulation of RBC in Supplementary Fig. 9. It looks like that the authors used live RBCs rather than gelated RBCs. The authors need to address whether GMs had the decoy and lung targeted effects in therapeutic models compared to gelated RBCs. In Figure 7 A and B, Cy5 and Cy5+ GMs are not good controls to address the

lung targeting of GMs.

2. The new data did not support the drawing in Figure 1C because the authors did not have the evidence to show that GMs transmigrated in the lung.

3. The short circulation time of GMs suggested that GMs lacked the decoy effect.

4. The supplementary Fig. 4 shows that GMs did not dramatically improve the decoy effect compared to membrane nanovesicles of macrophages. In the CIA model, GMs were locally injected. The authors need to demonstrate GMs were better than nanovesicles in the CIA model.

Reviewer #3 (Remarks to the Author):

Thank you to the authors for a very comprehensive revision that to a large extent has answered all my previous concerns. Two minor comments:

1. The IC50 values from Figure 3 mentioned in the text on page 7 should also be updated to cells/ml instead of ug/ml

2. The flow data added in Sup.fig 10 looks uncompensated. This should be corrected so analysis is performed on compensated samples.

Response Letter

Reviewer #1 (Remarks to the Author):

Although the resubmitted manuscript adds a lot of data, there are still some questions that have not been seriously answered.

Response: Thanks for the valuable comments, and we have conducted additional experiments and added detailed description to address your further concerns one by one.

1. The swelling properties of hydrogels require further evaluation.

Authors only examined the swelling ratio of hydrogels, which does not account for swelling properties. Confusingly, the authors concluded that the maximum swelling ratio of the gel was achieved after 5 days, while the GM remained intact in the body for up to 5 days, which seems difficult to achieve.

Response: Thanks for the valuable comments. Hydrogels utilized for GMs construction were prepared from Phe-CS and CB[8] in PBS solution, in which the solvent (PBS solution) and the concentration of crosslinker (50 μM of CB[8]) were well defined. The swelling profile of hydrogels from different concentrations of Phe-CS in PBS solution was drawn by measuring the swelling ratios at predetermined timepoints. The equilibrium swelling degree (q_e) values were in the range of 120% - 340%. After the equilibrium swelling was reached in 5 days, all samples were kept in buffer solution for additional 3 days, and q_e values remained practically constant. The swelling properties of hydrogel depended on many factors such as network density, solvent nature and polymer-solvent interaction parameter. As shown in **Supplementary Fig. 1C**, when the concentration of the crosslinker (CB[8]) was overdosed (e.g. 50 μM), the hydrogel swelling degree decreased with the increasing concentration of Phe-CS. This is because when the concentration of chitosan increases, the hydrogen bond associated structure would easily form between side chains of chitosan, which may function as crosslinking point in the crosslinking network structure.

Of note, these hydrogels needed 5 days to reach the equilibrium swelling state, but maximum swelling ratio of hydrogel was achieved from lyophilized hydrogel. Actually, in practice, lyophilized GMs were firstly dissolved in PBS solution and reached an equilibrium swelling state before utilization for experimental studies, or freshly prepared GMs could be immediately used. Herein, freshly prepared hydrogel from 2% (wt%) of Phe-CS and 50 μM of CB[8], with the same concentration utilized to prepare GMs, was further utilized to determine the equilibrium swelling time and ratio. A low equilibrium swelling ratio of ~16% was quickly reached in 3 days when stored in PBS solution (**Supplementary Fig. 1D**). This result indicated that freshly prepared hydrogel almost had no volume and mass change, contributing to a stable and intact structure of GMs in physiological environment.

Supplementary Fig. 1C. Swelling profiles of hydrogels prepared from different concentrations of Phe-CS (0.1%, 0.5%, 1%, 2% and 5%, wt%) and 50 μ M of CB[8] in PBS solution at 37 $^{\circ}$ C.

Supplementary Fig. 1D. Swelling profile of freshly prepared hydrogel from 2% (wt%) of Phe-CS and 50 μ M of CB[8] in PBS solution at 37 $^{\circ}$ C.

2. Authors suggested that the drug loading capacity of DS-GMs used for anti-arthritis treatment was \sim 0.4%. In the treatment of acute pneumonia, the weight content of QT in QT-GMs was \sim 1.2%. It is amazing that such a low drug load can achieve such good results, so the relevant mechanism research must be performed. In addition, is my concern that such a good treatment effect has similar effects in other inflammatory models?

Response: Thanks for the valuable comments, and we should calculate the drug loading capacity of GMs based on the lyophilized DS-GMs or QT-GMs, rather than the freshly prepared DS-GMs or QT-GMs. Freshly prepared DS-GMs or QT-GMs was almost on a swelling equilibrium state (**Supplementary Fig. 1D**) and full of PBS solution. In addition, the large molecular weight of hydrogel materials (CB[8] and Phe-CS) led to a big intracellular hydrogel mass. These two factors contributed to a high mass of \sim 2.74 ng per swelling GM and relative low drug loading capacity per previous calculation. Thus, to make the data more intuitive and avoid confusion, we determined the drug loading capacity of lyophilized product. After lyophilization, the drug loading capacity of DS-GMs and QT-GMs was calculated and updated to be 1.4% and 4.2%, respectively (**Page 25**). For in vivo treatment, AP mice were *i.v.* administered with 3×10^6 of QT-GMs per mouse, and the dose of QT reached 5 mg kg^{-1} . CIA rats were locally administered with 3×10^6 of DS-GMs per rat, and the dose of DS reached 1 mg kg^{-1} . The administration dosages of QT and DS were within or slightly lower than the

common dose range utilized to treat mice pneumonia (*Microbial Pathogenesis*, 2020, 140, 103934) and rat arthritis (*Journal of Ethnopharmacology*, 2020, 250, 112435 & *Biomedical Reports*, 2017, 7(2), 179-182.). Actually, the drug loading capacity of GMs could be regulated by changing the drug concentration of DS or QT in Phe-CS solution before infiltration into macrophage under freezing condition.

3. In vivo pharmacokinetic studies need to be provided.

Response: To conduct in vivo pharmacokinetic study on GMs, CytoTrace™ Red CFDA, a red fluorescent cell staining dye, was selected to label whole GMs via reactions with cellular components to form cell-impermeant products. AP mice were i.v. injected with 5×10^6 of Red CFDA-labelled GMs (n=3), and the organs (heart, liver, spleen, lungs and kidneys) were collected for ex vivo fluorescent imaging after administration for different durations (1, 6, 12, 24, 48 and 72 h). All mice showed bright red fluorescence in the lung immediately after administration, and the fluorescence intensity quickly decreased but still maintained at a certain level for 48 h (**Supplementary Fig. 9C**). After administration for 6 h, the spleen started to show fluorescence signal and quickly increased at beginning, but eventually disappeared post administration for 48 h. The liver intensity followed a similar pattern of the fluorescence changes of spleen, and continuously increased in 48 h post administration but finally decreased at 72 h. Together with a large area under fluorescence curve (AUC) in the lung, spleen and liver (**Supplementary Fig. 9D**), these results indicated that GMs had a good response to accumulate in the inflammatory lung tissue, and then may get phagocytosed by monocyte phagocytic system in the spleen to break down. Finally, GMs debris that could not be reutilized was transferred to the liver, and metabolized to produce water-soluble substances and excreted with urine, or directly excreted with feces to the intestinal tract.

Blood was collected from the other batch of mice at pre-determined timepoints (0.1, 0.5, 1, 2, 3, 4.5, 6, 12, and 18 h). As shown in **Supplementary Fig. 9E**, the blood fluorescent possessed highest intensity after administration for 0.1 h, and quickly decreased at the beginning. Then the fluorescence signal slowly decreased and almost disappeared after administration for 18 h. According to the fluorescence intensity-time curve (**Supplementary Fig. 9F**), GMs exhibited a long half-life of drug clearance ($t_{1/2} = 5.6$ h), indicating sufficient circulation time for GMs to neutralize systemic proinflammatory cytokines, due to the intact cell morphology and membrane structure of GMs to avoid fast clearance by monocyte phagocytic system.

Supplementary Fig. 9. Ex vivo biodistribution of Cy5-RBCs and in vivo pharmacokinetics of GMs in AP mice. (C and D) AP mice were i.v. injected with 5×10^6 of Red CFDA-labelled GMs ($n=3$). The organs (heart, liver, spleen, lungs and kidneys) were collected for ex vivo fluorescent imaging after administration for different times (1, 6, 12, 24, 48 and 72 h) (C), and the change of fluorescence intensity was quantified by IVIS (D). (E and F) Blood was collected from the other batch of mice for ex vivo fluorescent imaging at pre-determined timepoints (0.1, 0.5, 1, 2, 3, 4.5, 6, 12, and 18 h) (E), and the change of fluorescence intensity was quantified by IVIS (F). All data was presented as mean \pm s.d. ($n = 3$).

4. Notably, the assessment of macrophages is overly simplistic given the highly controversial nature of M1 and M2 classification (as well as the M1/M2 binary relevance to in vivo function, see <https://doi.org/10.1186/s12885-018-4457-8> and <https://doi.org/10.12703/P6-13> and <https://doi.org/10.1038/nri3088> and <https://doi.org/10.1016/j.addr.2017.05.010>). Further, with regards to the effect on the macrophage population, it is unclear if the effects are due to polarization of the macrophage cells or deletion of immunosuppressive phenotypes.

Response: Thanks for the suggestion. According to the literatures provided by reviewers, the use of terms M1 and M2 remains controversial because of the lack of tightly defined criteria to score phenotypes. In general, mixed phenotypes or populations with different phenotypes coexist, indicating that the M1/M2 classification is too simplistic for this transcriptionally dynamic cell type. However, the proinflammatory macrophage phenotype retains communication value as long as the limitation of this classification is considered and, herein, it is used to reflect the usage

in this paper. The term M1 macrophage had been revised into proinflammatory macrophage (F4/80⁺CD11b⁺CD86⁺ cells) in manuscript. Proinflammatory macrophage could be induced in vitro by bacteria product (LPS) (<https://doi.org/10.12703/P6-13>), and immunofluorescence staining in lung section showed that LPS treatment induced the significant filtration of F4/80⁺ cells (one marker of macrophage) and locally polarized macrophage into proinflammatory phenotype (F4/80⁺CD86⁺ cells) (**Fig. 7J**), which was further quantified by the increased percent of F4/80⁺CD11b⁺CD86⁺ cells (**Fig. 7K, 7L** and **Supplementary Fig. 10**). Due to the neutralization effect towards proinflammatory cytokine and bacteria product (LPS), GMs treatment reduced the infiltration ratio of F4/80⁺CD11b⁺CD86⁺ cells in pneumonia mice, indicating macrophage polarization into proinflammatory phenotype was ameliorated. In addition, QT was reported to induce inhibitory effects on inflammatory macrophage polarization via the NF-κB and IRF5 signaling (Biochemical Pharmacology, 2018, 154, 203-212), which was also confirmed by the decreased lung infiltration of F4/80⁺CD11b⁺CD86⁺ cells in **Fig. 7K, 7L** and **Supplementary Fig. 10**. Subsequently, the combination therapy of QT-GMs exhibited the most pronounced therapeutic outcome in decreasing F4/80⁺CD11b⁺ cells infiltration and efficiently inhibited macrophage polarization into proinflammatory phenotype, when compared with other therapeutic groups. The effect of QT-GMs treatment on macrophage population was mainly beneficial from cytokine or LPS neutralization effect of GMs and anti-inflammation effect of QT.

Reviewer #2 (Remarks to the Author):

The authors provided some new data to address the reviewer's questions, but there are still some concerns.

Response: Thanks for the kind review and valuable comments on our article. Accordingly, we have supplemented additional experiments and discussion to provide more solid evidences to address the reviewer's concerns.

1. The authors compared the lung accumulation of RBC in Supplementary Fig. 9. It looks like that the authors used live RBCs rather than gelated RBCs. The authors need to address whether GMs had the decoy and lung targeted effects in therapeutic models compared to gelated RBCs. In Figure 7 A and B, Cy5 and Cy5⁺ GMs are not good controls to address the lung targeting of GMs.

Response: Sorry for our previous misunderstanding. To precisely follow this advice, accordingly, we prepared gelated RBCs (GRBCs) as a non-targeting control group, to evaluate the biodistribution in pneumonia mice in comparison to GMs. To track their in vivo biodistribution, RBCs and macrophages were both stained with membrane red fluorescence dye (DiD) before intracellular gelation. AP mice (n = 3) were i.v. injected with DiD stained GMs (DiD-GMs) and GRBCs (DiD-GRBCs), respectively, at same fluorescence intensity for ex vivo fluorescent imaging (**Supplementary Fig. 9A** and

9B), which referred to as 5×10^6 of DiD-GMs and 2×10^7 of DiD-GRBCs. After injection for 6 h, DiD-GRBCs treated mice exhibited nearly no fluorescence intensity in the lung tissue, and DiD-GRBCs were mainly distributed in the liver. In contrast, a bright fluorescent intensity was observed in the DiD-GMs treated mice, suggesting the inflammatory tropism of GMs.

Supplementary Fig. 9A and 9B. AP mice ($n = 3$) were i.v. administered with DiD-GMs and DiD-GRBCs at the same fluorescence intensity, which referred to 5×10^6 of DiD stained GMs and 2×10^7 of DiD stained GRBCs, respectively. After injection for 10 min, different organs (heart, liver, spleen, lungs and kidneys) were collected for ex vivo fluorescence imaging (A), and the fluorescence intensity was quantified by IVIS (B).

2. The new data did not support the drawing in Figure 1C because the authors did not have the evidence to show that GMs transmigrated in the lung.

Response: Thanks for the comments. Accordingly, we have redrawn the scheme of **Fig. 1C**. Considering that GMs are relatively rigid and do not have the same level of plasticity of living macrophages, it is indeed hard to penetrate through lung vasculature, and most of them are likely located inside the abundant lung capillaries to neutralize cytokines released from inflammatory pneumocyte. Thus, GMs are drawn only inside the lung capillaries rather than transmigration into the lung in the updated **Fig. 1C**.

Fig. 1. Schematic illustration of GMs construction for targeted anti-inflammatory treatment.

3. The short circulation time of GMs suggested that GMs lacked the decoy effect.

Response: In previous detection of GM circulation, transportation and retention in pneumonia mice, GMs labelled with red fluorescent membrane dye (DiD) were utilized for fluorescence imaging. However, nearly no fluorescence signal was detected in the blood even after administration for 2 h, and we thought that this should be attributed to the small fluorescent membrane area and weak fluorescence intensity of DiD labelled GMs. Thus, to better track GM's fate, CytoTrace™ Red CFDA, a red fluorescent cell staining dye, was selected to label whole GMs via reactions with cellular components (including membrane, cytoplasm and nucleus) to form cell-impermeant products. AP mice were *i.v.* injected with Red CFDA-labelled GMs at a number of 5×10^6 ($n = 3$), and blood samples were collected at pre-determined timepoints (0.1, 0.5, 1, 2, 3, 4.5, 6, 12, and 18 h) after administration. As shown in **Supplementary Fig. 9E**, the blood fluorescence possessed the highest intensity after administration for 0.1 h, and moderately decreased at the beginning. Then the fluorescence signal slowly decreased and nearly disappeared after administration for 18 h. According to the fluorescence intensity-time curve (**Supplementary Fig. 9F**), GMs exhibited a long half-life of drug

clearance ($t_{1/2} = 5.6$ h), indicating sufficient circulation time for GMs to neutralize systemic proinflammatory cytokines. This result was mainly due to the intact cell morphology and membrane structure of GMs to avoid clearance by monocyte phagocytic system.

Supplementary Fig. 9E and 9F. AP mice were i.v. injected with Red CFDA-labelled GMs at a number of 5×10^6 ($n = 3$). Blood was collected from the other batch of mice for ex vivo fluorescent imaging after administration for different times (0.1, 0.5, 1, 2, 3, 4.5, 6, 12, and 18 h) (E), and the change of fluorescence intensity was quantified by IVIS (F).

4. The supplementary Fig. 4 shows that GMs did not dramatically improve the decoy effect compared to membrane nanovesicles of macrophages. In the CIA model, GMs were locally injected. The authors need to demonstrate GMs were better than nanovesicles in the CIA model.

Response: Thanks for the suggestion. Accordingly, we have supplemented experiments to compare the decoy efficacy between GMs and macrophage membrane coated nanoparticles (MM-NPs) in both in vitro cytokine sequestration and in vivo treatment on CIA rats. For in vivo evaluation, CIA rat treated with MM-NPs prepared from same number of macrophages (3×10^6), was supplemented as a control group. As shown in **Supplementary Fig. 5** and **Fig. 6**, MM-NPs exhibited moderate cytokine sequestration effect and could alleviate arthritis symptom to a certain extent. The therapeutic efficacy of MM-NPs was comparable to that of α TNF- α , but GMs still exhibited a much better therapeutic effect on CIA rat with statistical difference, when compared with MM-NPs. This should be attributed to the membrane protein loss and spatial disorder during MM-NP preparation process, leading to a relative lower cytokine neutralization efficiency. The intact membrane structure of GMs contributed to a more effective anti-arthritis outcome than that of MM-NPs (**Page 14**).

Supplementary Fig. 5. Photographs of the rat paws taken at all time points in different treatment groups during therapeutic treatment. The swelling paws of CIA rats after 4 weeks of arthritis development were injected with saline, DS, GMs, DS-GMs, aTNF- α , and DS+ aTNF- α at 3×10^6 of GMs per rat, 1 mg kg^{-1} of DS and 1 mg kg^{-1} of aTNF- α . Rats treated with 2 mg kg^{-1} of hydrogel, 3×10^6 of MAs and 2 mg kg^{-1} of PSL, and MM-NPs prepared from 3×10^6 of MAs served as control groups. After treatment once every four days until Day 52, representative photos at pre-determined time points demonstrated the progression of mouse paw inflammation.

Supplementary Fig. 6. Therapeutic efficacy of hydrogel, MAs, DS+aTNF- α and GC in CIA rats. (A) The changes of paw thickness in CIA rats after treatments with DS+ aTNF- α at 1 mg kg⁻¹ of DS and 1 mg kg⁻¹ of aTNF- α , 2 mg kg⁻¹ of hydrogel, 3x10⁶ of MAs, MM-NPs prepared from 3x10⁶ of MAs, and 2 mg kg⁻¹ of PSL. (B and C) Representative images of Fibronectin staining (B) and CD248 staining (C) on the on the synovial intimal lining from treated rats. Scale bar: 100 μ m. (D) The concentrations of serum TNF- α and IL-1 β at the endpoint of treatment. (E) The ankle diameter changes of treated rats compared to that of the normal rats recorded at Day 52. (F) The overall arthritis score of the treated rats recorded every four days with a total of 28 days after arthritis induction. All data was presented as mean \pm s.d. (n = 3). All statistical analyses were conducted by using One-Way ANOVA. *P \leq 0.05 and ***P \leq 0.001.

Reviewer #3 (Remarks to the Author):

Thank you to the authors for a very comprehensive revision that to a large extent has answered all my previous concerns. Two minor comments:

Response: Thanks for your kind recognition of our previous efforts in improving the

overall quality of the manuscript, and for the kind recommendation for publication upon minor revisions.

1. The IC₅₀ values from Figure 3 mentioned in the text on page 7 should also be updated to cells/ml instead of ug/ml

Response: Thanks. Accordingly, the description on the IC₅₀ values from **Fig. 3** has been revised as cell number/mL. “The IC₅₀ (half maximal inhibitory concentration) value of GMs was determined to be 5.2×10^5 cells mL⁻¹ for TNF- α and 2.7×10^5 cells mL⁻¹ for IL-1 β , respectively” (**Page 7**).

2. The flow data added in Sup.fig 10 looks uncompensated. This should be corrected so analysis is performed on compensated samples.

Response: Thanks for your suggestion, and the flow cytometry result in **Supplementary Fig. 10** has been supplemented with compensation. The updated result is shown as follows (**next page**), and we also updated in the supplementary document.

Supplementary Fig. 10. Gating strategy of flow cytometry analysis on the filtration of macrophages ($F4/80^+CD11b^+$ cells) and M1 macrophages ($F4/80^+CD11b^+CD86^+$ cells) in lung tissues. AP mice were blindly and randomly divided into four groups ($n = 3$), and i.v. administered with saline, QT, QT-GMs, and GMs at 5×10^6 of GMs per mouse, 5 mg kg^{-1} of QT, 2 mg kg^{-1} of hydrogel and 3×10^6 of MAs. After administration for 6 h, the lung tissues were collected for flow cytometry analysis.

REVIEWER COMMENTS

Reviewer #1 (Remarks to the Author):

Thanks to the authors for making the revisions to address some of my previous concerns, however, for the M1/M2 classification I mentioned earlier, it is recommended to use more accurate markers for the assessment of TAM polarization. Besides, the measurement of swelling may be misleading due to secondary effects, e.g. changes in ionic strength of the buffer. To justify your idea, please provide detailed FCS or FCCS analytics in serum to reveal its stability. Finally, the authors should consider the potential autoimmune responses due to the nanosystem's increased liver accumulation.

Reviewer #2 (Remarks to the Author):

Gao et al revised their manuscript "Intracellularly gelated macrophages for targeted therapy of inflammatory diseases". The revision did not address the reviewer's questions, and the controls of RBC were not performed in the therapy studies, and proper studies.

Targeted lung of macrophages does not really depend on inflamed vasculature, but is dependent on the contraction of lung capillaries. Red blood cells travel through lung capillaries through their shape changes, thus gelated RBC should be stuck in the lung capillaries. However, the new results provided by the authors showed that gelated RBC was accumulated in the liver. RBCs have a longer circulation time than cells, thus supplementary Figure 9 is questionable.

Lacking of proper controls and comparable studies in the manuscript is hard to make a conclusion on the novelty of gelated macrophage systems. I suggest that it is rejected.

Response Letter

(Note that **Reviewer #3** recommended for publication already during the last round of review)

Reviewer #1 (Remarks to the Author):

Thanks to the authors for making the revisions to address some of my previous concerns, however, for the M1/M2 classification I mentioned earlier, it is recommended to use more accurate markers for the assessment of TAM polarization. Besides, the measurement of swelling may be misleading due to secondary effects, e.g. changes in ionic strength of the buffer. To justify your idea, please provide detailed FCS or FCCS analytics in serum to reveal its stability. Finally, the authors should consider the potential autoimmune responses due to the nanosystem's increased liver accumulation.

Response: Thanks for the careful review and valuable comments. According to reviewer's suggestion, macrophage was labelled with two markers of CD80 and CD86 to assess the M1-like polarization more accurately. As shown in new **Fig. 2K**, CD80⁺CD86⁺ cells were observed as proinflammatory macrophage, and all of MAs was polarized into proinflammatory type after incubation in 10 μ M of TNF- α for 12 h. In contrast, nearly no CD80⁺CD86⁺ cells were detected in TNF- α treated GMs, indicating GMs could not be activated in inflammatory microenvironment.

In addition, to investigate the TAM polarization in inflammatory lung, CD80⁺CD86⁺ cells were also evaluated as M1-like cells (**Fig. 7L** and **Supplementary Fig. 10**). GMs treatment reduced the infiltration ratio of F4/80⁺CD80⁺CD86⁺ cells in pneumonia mice, indicating macrophage polarization into proinflammatory phenotype was ameliorated.

To investigate the hydrogel stability in serum, Phe-CS was conjugated with FITC to label hydrogel for fluorescence correlation spectroscopy (FCS) analysis. In comparison to the emission curve of FITC conjugated Phe-CS solution at an excitation wavelength of 480 nm, hydrogel exhibited a red shift from 492 nm to 574 nm (**Supplementary Fig. 2C**), confirming successful hydrogel formation and labelling. After storage in serum for different durations (0, 1, 3, 5, 7 days), only a minor or nearly negligible blue shift was observed in hydrogel emission curve, indicating high stability of freshly prepared hydrogel. Furthermore, plot scattering of GMs, analyzed by flow cytometry (**Supplementary Fig. 2D**), exhibited similar FSC-SSC distribution even after storage in serum for 7 days, suggesting that the size and granularity of GMs did not change. The results further confirmed the high stability of GMs.

For concerns about the potential autoimmune response, GMs were constructed from autologous macrophage and natural hydrogel, and both components show a high biocompatibility. In vivo biosafety evaluation in our study further suggested a good safety profile of GMs, and no damage on different organs and systemic inflammation were observed in vivo, including liver histological analysis and liver function measurement (**Supplementary Fig. 13**), although a large number of GMs were

transferred to liver for metabolism. Nevertheless, long-term administration of GMs might induce potential liver toxicity or autoimmune response. The discussion about the potential autoimmune response was supplemented in the manuscript to make readers aware (Page 22).

Supplementary Fig. 10. Gating strategy of flow cytometry analysis on the filtration of polarized macrophages ($F4/80^+CD80^+CD86^+$ cells) in lung tissues.

Fig. 2. Preparation and in vitro assessment of GMs. (K) The ratio of proinflammatory MAs ($CD80^+CD86^+$ cells) was detected in MAs and GMs after treatment with $TNF-\alpha$ for 12 h by flow cytometry.

Supplementary Fig. 2. Characterization of Phe-CS based hydrogel. (C) Phe-CS was conjugated with FITC to label hydrogel for fluorescence correlation spectroscopy (FCS) analysis. After storage in serum for different durations (0, 1, 3, 5, 7 days), the fluorescence emission spectrum of hydrogel was determined at an excitation wavelength of 480 nm. **(D)** Plot scattering of GMs after storage in serum for 7 days.

Reviewer #2 (Remarks to the Author):

Gao et al revised their manuscript “Intracellularly gelled macrophages for targeted therapy of inflammatory diseases”. The revision did not address the reviewer’s questions, and the controls of RBC were not performed in the therapy studies, and proper studies. Targeted lung of macrophages does not really depend on inflamed vasculature, but is dependent on the contraction of lung capillaries. Red blood cells travel through lung capillaries through their shape changes, thus gelled RBC should be stuck in the lung capillaries. However, the new results provided by the authors showed that gelled RBS was accumulated in the liver. RBCs have a longer circulation time than cells, thus supplementary Figure 9 is questionable.

Lacking of proper controls and comparable studies in the manuscript is hard to make a conclusion on the novelty of gelled macrophage systems. I suggest that it is rejected.

Response: Thanks for the comments. We actually supplemented gelated RBCs (GRBCs) as a control group to verify the cytokine neutralization effect of GMs for in vitro and in vivo anti-inflammation therapy. As shown in **Supplementary Fig. 4A** and **4B**, 3×10^6 of GMs and GRBCs were incubated in 1 mL of PBS solution containing 10 ng mL⁻¹ of TNF- α and IL-1 β , respectively. After incubation for 1 h, GMs significantly reduced the TNF- α and IL-1 β level into 2.93 and 2.36 $\mu\text{g mL}^{-1}$, respectively, in supernatant. In contrast, GRBCs showed negligible effects on the cytokine concentration. These results exhibited a strong cytokine sequestration ability of GMs in comparison to GRBCs. Furthermore, in acute pneumonia mice, the inflammatory symptoms of lung were not alleviated after GRBCs treatment (**Supplementary Fig. 10, 11** and **12**), comparable to the disease model group. In contrast, GMs efficiently reduced the infiltration of immune cells and improved histological characterization in lung. These results demonstrated the ability of GMs to neutralize cytokine conferred by their intact macrophage membrane rather than non-immune cell membrane such as RBC. Collectively, these comparison study with GRBCs further demonstrated the novelty of our gelated macrophage system in maintaining intact membrane structure, including protein type, content and spatial distribution, for comprehensive neutralization therapy on inflammatory diseases.

For in vivo biodistribution of RGBCs, we also speculated that GRBCs would accumulate in lung tissue because of the abundant lung capillaries and non-viability of GRBCs. However, all ex vivo fluorescent imaging on three mice did show a bright fluorescent intensity in liver tissues after injection with DiD stained GRBCs (DiD-GRBCs), and exhibited nearly no fluorescence intensity in lung tissues (**Supplementary Fig. 9A**). It was revealed in previous study that the liver relies on a buffer system consisting of bone marrow-derived monocytes, which consume damaged RBCs in the blood and settle in the liver (*Nature Medicine*, 2016, 22, 945-951). Thus, the liver accumulation might be attributed to the monocyte phagocytosis towards damaged GRBCs and further hitchhiking delivery to liver. In contrast, DiD-GMs exhibited a quick accumulation in inflammatory lung. Together with the evidences of in vitro static and dynamic binding of GMs to inflammatory cells (**Fig. 4**) and in vivo accumulation in the inflammatory footpad of CIA rats (**Fig. 5A** and **5B**), these results collectively confirmed the inflammatory tropism effect of GMs.

Supplementary Fig. 4. Cytokine neutralization efficiency of GMs, MM-NPs and GRBCs. (A and B) GMs and MM-NPs prepared from same number of macrophages (3×10^6), and 3×10^6 of GRBCs were treated with 10 ng mL⁻¹ of TNF- α (A) and IL-1 β (B) for 1 h. The supernatant cytokine concentrations were measured by Elisa kits.

Supplementary Fig. 11. Therapeutic efficacy of hydrogel, MAs and GRBCs on acute lung inflammation in AP mice. (A) AP mice were i.v. administered with 2 mg kg⁻¹ of hydrogel and 3x10⁶ of MAs and GRBCs, respectively. After administration for 6 h, the lung tissues were collected for further analysis.

Supplementary Fig. 9. Ex vivo biodistribution of DiD-GMs and DiD-GRBCs after administration in AP mice for 6 h.

REVIEWER COMMENTS

Reviewer #1 (Remarks to the Author):

I suggest that this manuscript be rejected for the following reasons:

Firstly, the assessment of TAMs is overly simplistic given the highly controversial nature of M1 and M2 classification (as well as the M1/M2 binary relevance to in vivo function) within immunology circles. (see <https://doi.org/10.1186/s12885-018-4457-8> and <https://doi.org/10.12703/P6-13> and <https://doi.org/10.1038/nri3088> and <https://doi.org/10.1016/j.addr.2017.05.010> highlight the extensive discussion on this topic). Therefore, I have always suggested using more accurate markers to assess TAM polarization, but the authors have never done so. I am not suggesting the authors necessarily repeat these studies with more robust methods like scRNA sequencing, but I am concerned that the mechanistic conclusions aren't well supported. Further, the authors had to consider the potential autoimmune response caused by increased liver accumulation by nanosystems, rather than simply examining liver toxicity. It's hard to be convincing.

Response to Review

Reviewer #1 (Remarks to the Author):

Firstly, the assessment of TAMs is overly simplistic given the highly controversial nature of M1 and M2 classification (as well as the M1/M2 binary relevance to in vivo function) within immunology circles. (see <https://doi.org/10.1186/s12885-018-4457-8> and <https://doi.org/10.12703/P6-13> and <https://doi.org/10.1038/nri3088> and <https://doi.org/10.1016/j.addr.2017.05.010> highlight the extensive discussion on this topic). Therefore, I have always suggested using more accurate markers to assess TAM polarization, but the authors have never done so. I am not suggesting the authors necessarily repeat these studies with more robust methods like scRNA sequencing, but I am concerned that the mechanistic conclusions aren't well supported.

Further, the authors had to consider the potential autoimmune response caused by increased liver accumulation by nanosystems, rather than simply examining liver toxicity. It's hard to be convincing.

Response: Sorry that the accurate assessment of TAMs was not properly addressed during the previous round of revision (from which you recommended acceptance upon revisions), because this aspect was not the focus of the study, and the biomarkers we previously used, although simple, are common ones to identify TAM in the area of biomaterials (as shown in numerous previous publications). In addition, macrophage polarization is a dynamic process. Under certain conditions, M1 and M2 can also transform into each other, and this transformation enables macrophages to maintain tissue homeostasis and body balance when the microenvironment changes. There is no absolute opposition between the two phenotypes of M1 and M2. They are not mutually exclusive, instead they often coexist.

Nevertheless, with our good faith to the reviewer and to fully address this reviewer's final suggestion and satisfy his (or her) request, more accurate markers (CD86, CD40 and CD80) of proinflammatory phenotype were selected to assess in vitro and in vivo TAM polarization, and to obtain a more comprehensive result. As shown in new **Fig. 2K** and **Supplementary Fig. 2E**, after incubation in 10 μ M of TNF- α for 12 h, the ratios of CD86⁺ macrophage, CD40⁺ macrophage and CD80⁺ macrophage increased, indicating proinflammatory polarization of macrophage after TNF- α treatment. In contrast, TNF- α treated GMs exhibited negligible CD86⁺, CD40⁺ and CD80⁺ macrophages, comparable to unpolarized MAs. These results suggest that GMs could not be activated in inflammatory microenvironment. In addition, to investigate the TAM polarization in inflammatory lung, CD86, CD40 and CD80 were also selected as the key biomarkers for comprehensive evaluation (**Supplementary Fig. 10**, **Fig. 7L** and **Supplementary Fig. 11G**). Flow cytometry analysis exhibited the increased ratios of CD86⁺, CD40⁺ and CD80⁺ cells, respectively, in F4/80⁺CD11b⁺ cells in pneumonia mice, (**Fig. 7K**, **7L** and **Supplementary Fig. 10**), confirming that the LPS treatment locally polarized macrophage into proinflammatory phenotype. In contrast, the

infiltration ratios of CD86⁺, CD40⁺ and CD80⁺ macrophages were reduced after GMs treatment, indicating that macrophage polarization into proinflammatory phenotype was ameliorated.

For additional concerns about the potential autoimmune response caused by increased liver accumulation of GMs, on the top of previous safety evaluation, we further evaluated the autoimmune characterizations in the liver from mice treated with GMs for 14 days. Firstly, the infiltration of CD4⁺ T and CD8⁺ T cells in mouse liver was measured to investigate the T-cell-mediated autoimmune response. As shown in the representative immunofluorescence images (**Supplementary Fig. 14A**), only a few CD4⁺ T cells and CD8⁺ T cells infiltrated into the liver after GMs treatment, which was similar to the level of liver from the healthy mice. At the same time, the mouse liver showed a similar appearance to the normal liver from healthy mice as well, without any symptoms of inflammation (**Supplementary Fig. 14B**). HE staining analysis showed negligible infiltration of inflammatory cells (**Supplementary Fig. 14C**). Furthermore, considering that fibrosis is one of the main features of terminal autoimmune hepatitis, we used Sirius red staining to detect the fibrosis level on liver. As shown in **Supplementary Fig. 14D**, GM treated mice almost exhibited no fluorescence of collagenous fiber, comparable to normal mice. We also used immunofluorescence analysis to detect the expression of α -SMA in the mouse liver (**Supplementary Fig. 14E**). α -smooth muscle actin (α -SMA) is normally expressed around the blood vessels, and its levels also did not increase in GM treated mouse liver. Together with the result of normal histological feature by HE staining (**Supplementary Fig. 13A**) and liver function markers (ALT and AST) (**Supplementary Fig. 13D**), our data suggested that GM treatment would not induce liver autoimmune response, which should be attributed to the quick clearance and quite low accumulation in liver after *i.v.* treatment for 3 days (**Supplementary Fig. 9C** and **9D**). This result about autoimmune response was supplemented in the manuscript to make readers aware (**Page 22** and **23**).

Fig. 2K. The ratio of proinflammatory polarization (CD86⁺ cells, CD40⁺ cells and CD80⁺ cells) was detected in MAs and GMs after treatment with TNF- α for 12 h by flow cytometry.

Supplementary Fig. 2E. Quantitative results of the proinflammatory polarization of MAs and GMs (CD86⁺ cell, CD40⁺ cell and CD80⁺ cell) after treatment with 10 μ M of TNF- α for 12 h.

Supplementary Fig. 10. Gating strategy of flow cytometry analysis on the filtration of macrophages (F4/80⁺CD11b⁺ cells) and proinflammation polarized macrophages (the ratio of CD86⁺ Cells, CD40⁺ cells and CD80⁺ cells in F4/80⁺CD11b⁺ cells) in lung tissues. AP mice were blindly and randomly divided into four groups (n=3), and *i.v.* administered with saline, QT, QT-GMs, GMs and EGMs at 3x10⁶ of GMs per mouse, 5 mg kg⁻¹ of QT, 2 mg kg⁻¹ of hydrogel, 3x10⁶ of MAs and 3x10⁶ of GRBCs. After administration for 6 h, the lung tissues were collected for flow cytometry analysis.

Fig. 7K and 7L. Flow cytometry analysis on the filtration of F4/80⁺CD11b⁺ cells and the ratio of CD86⁺ cells, CD40⁺ cells, CD80⁺ cells among them in the lungs of treated mice.

Supplementary Fig. 11F and 11G. Flow cytometry analysis on the filtration of F4/80⁺CD11b⁺ cells and the ratio of CD86⁺ cells, CD40⁺ cells, CD80⁺ cells among them in the lungs of treated mice. All data was presented as mean ± s.d. (n = 3). All statistical analyses were conducted using One-Way ANOVA.

Supplementary Fig. 14. Evaluation of autoimmune response in liver from GMs treated mice. (A) C57BL/6 mice were i.v. administered with 5×10^6 of DiD-GMs for different times (7 days and 14 days). Immunofluorescent staining of CD4 (blue) and CD8 (red) was conducted in liver sections. Scale bar: 100 μ m. (B) Liver appearance after treatment for 7 days and 14 days. (C) HE staining on mouse liver sections after treatment for 7 days and 14 days. Scale bar: 100 μ m. (D) Sirius red staining on mouse liver sections after treatment for 7 days and 14 days. Scale bar: 100 μ m and 50 μ m ($\times 200$ and $\times 400$ magnification). (E) α -SMA staining on mouse liver sections after treatment for 7 days and 14 days. Scale bar: 50 μ m. The experiments were repeated for three times ($n = 3$) and data was presented as mean \pm s.d. All statistical analyses were conducted using One-Way ANOVA.

Reviewer #2 (Remarks to the Author):

Gao et al revised their manuscript “Intracellularly gelled macrophages for targeted therapy of inflammatory diseases”. The revision did not address the reviewer’s questions, and the controls of RBC were not performed in the therapy studies, and proper studies. Targeted lung of macrophages does not really depend on inflamed vasculature, but is dependent on the contraction of lung capillaries. Red blood cells travel through lung capillaries through their shape changes, thus gelled RBC should be stuck in the lung capillaries. However, the new results provided by the authors showed that gelled RBS was accumulated in the liver. RBCs have a longer circulation time than cells, thus supplementary Figure 9 is questionable.

Lacking of proper controls and comparable studies in the manuscript is hard to make a conclusion on the novelty of gelled macrophage systems. I suggest that it is rejected.

Response: Thanks for the valuable comments. Accordingly, we supplemented gelled red blood cells (GRBCs) as a stiffer control and GMs derived from enzymes (pepsin and trypsin) treated macrophages (EGMs) as a non-targeting control group, to verify the cytokine neutralization effect of GMs for in vitro and in vivo anti-inflammation therapy. As shown in **Supplementary Fig. 4A** and **4B**, 3×10^6 of GMs, EGMs and GRBCs were incubated in 1 mL of PBS solution containing 10 ng mL^{-1} of TNF- α and IL-1 β , respectively. After incubation for 1 h, GMs significantly reduced the TNF- α and IL-1 β level into 2.93 and $2.36 \mu\text{g mL}^{-1}$, respectively, in supernatant. In contrast, GRBCs and EGMs showed negligible effects on the cytokine concentration. These results exhibited a strong cytokine sequestration ability of GMs in comparison to GRBCs and EGMs. Furthermore, in acute pneumonia mice, DiD stained GRBCs (DiD-GRBCs) and DiD stained EGMs (DiD-EGMs) exhibited nearly no fluorescence intensity in the lung tissue after *i.v.* injection for 6 h, and they were mainly distributed in the liver (**Supplementary Fig. 9A** and **9B**). In contrast, a bright fluorescent intensity was observed in the lungs of mice treated with DiD stained GMs (DiD-GMs), suggesting that GMs exhibited significant inflammatory tropism in comparison to GRBCs and EGMs. This result indicated that macrophage membrane protein played a key role in the targeted delivery to inflammatory lung.

Finally, in the treatment of pneumonia mice, the inflammatory symptoms of lung were not alleviated after GRBCs and EGMs treatment (**Supplementary Fig. 10**, **11** and **12**), comparable to the disease model group. In contrast, GMs efficiently reduced the infiltration of immune cells and improved histological characterization in lung. These results demonstrated the ability of GMs to neutralize cytokine was conferred by their intact macrophage membrane protein, rather than stiffer cells such as GRBCs or macrophage without membrane protein such as EGMs. Collectively, these comparison study with GRBCs and EGMs further demonstrated the novelty of our gelled macrophage system in maintaining intact membrane structure, including protein type, content and spatial distribution, for comprehensive neutralization therapy on inflammatory diseases.

For in vivo biodistribution of RGBCs, we also speculated that GRBCs would accumulate in the lung tissue because of the abundant lung capillaries and non-viability of GRBCs. However, all ex vivo fluorescent imaging on three mice did show a bright fluorescent intensity mainly in the liver tissues after injection with DiD stained GRBCs (DiD-GRBCs), and exhibited nearly negligible fluorescence intensity in the lung tissues (**Supplementary Fig. 9A and 9B**). This is not actually surprising, as it was revealed in previous study that the liver relies on a buffer system consisting of bone marrow-derived monocytes, which consume damaged RBCs in the blood and settle in the liver (*Nature Medicine*, 2016, 22, 945-951). Thus, the liver accumulation might be attributed to the monocyte phagocytosis towards damaged GRBCs and further hitchhiking delivery to liver. In contrast, DiD-GMs exhibited a quick accumulation in inflammatory lung. Together with the evidences of in vitro static and dynamic binding of GMs to inflammatory cells (**Fig. 4**) and in vivo accumulation in the inflammatory footpad of CIA rats (**Fig. 5A and 5B**), these results collectively confirmed the inflammatory tropism effect of GMs.

We have done our best to address all of the comments with the best faith in the reviewers, we hope your sincere support to this work, as the current set of data after four-round revisions have sufficiently demonstrated the concept and well supported the conclusions.

Supplementary Fig. 4. Cytokine neutralization efficiency of GMs, MM-NPs, GRBCs and EGMs. (A and B) GMs, MM-NPs and EGMs prepared from same number of macrophages (3×10^6), and 3×10^6 of GRBCs were incubated in 1 mL of PBS solution containing 10 ng mL^{-1} of TNF- α (A) and 10 ng mL^{-1} of IL-1 β (B) for 1 h, respectively. After removing the precipitates, the supernatant cytokine concentrations were measured by Elisa kits. (C) Membrane protein contents of GMs, MM-NPs and EGMs prepared from same number of macrophages (3×10^6) were analyzed by BCA protein assay. All data was presented as mean \pm s.d. (n = 3). All statistical analyses were conducted using One-Way ANOVA. * $P \leq 0.05$ and *** $P \leq 0.001$.

Supplementary Fig. 9. Ex vivo biodistribution and in vivo pharmacokinetics of GMs in AP mice. (A and B) AP mice (n=3) were *i.v.* administered with DiD-GMs, DiD-EGMs and DiD-GRBCs at the same fluorescence intensity, which referred to 5×10^6 of DiD stained GMs and EGMs, and 2×10^7 of DiD stained GRBCs, respectively. After injection for 6 h, different organs (heart, liver, spleen, lung and kidney) were collected for ex vivo fluorescence imaging (A), and the fluorescence intensity was quantified by IVIS (B).

Supplementary Fig. 11. Therapeutic efficacy of hydrogel, MAs and GRBCs on acute lung inflammation in AP mice. (A) AP mice were i.v. administered with 2 mg kg^{-1} of hydrogel, and 3×10^6 of MAs, EGMs and GRBCs, respectively. After administration for 6 h, the lung tissues were collected for HE staining and HO-1 staining. Scale bar: $100 \mu\text{m}$. (B) The wet/dry ratio of the collected lung tissues. (C) Serum levels of TNF- α and IL-1 β . (D) The concentrations of MPO and MDA in the lung tissues were analyzed by assay kits. (E) Fluorescence imaging on the filtration of F4/80⁺ cells in the lungs of treated mice. Scale bar: $50 \mu\text{m}$. (F and G) Flow cytometry analysis on the filtration of F4/80⁺CD11b⁺ cells and the ratio of CD86⁺ cells, CD40⁺ cells, CD80⁺ cells among them in the lungs of treated mice. All data was presented as mean \pm s.d. ($n = 3$). All statistical analyses were conducted using One-Way ANOVA.

Supplementary Fig. 12. Fluorescence imaging on the filtration of neutrophils (CD11b⁺Gr-1⁺ cells) in the lungs of treated mice, and semi-quantitative analysis by ImageJ software. Scale bar: 40 μ m.

REVIEWERS' COMMENTS

Reviewer #4 (Remarks to the Author):

Months ago, Gao et al. submitted a manuscript entitled “Intracellularly gelated macrophages for targeted therapy of inflammatory diseases” to Nature Communications. This manuscript underwent several rounds of evaluation, and I have been specifically asked by the editor to assess whether “I consider that the concerns of reviewer #2 have been addressed. This is particularly in terms of the control experiments asked for”.

In first round, the previous reviewer asked for missing controls: 1) the comparison of the behaviour of RBC with the one of gelated macrophages (GM); and 2) the controls of DS+anti-TNF (related to Fig 5). The authors have properly addressed point 2). Regarding point 1), they have compared the behaviour of GM with the one of non-gelated RBC. Therefore, in the second round, the reviewer logically asked for gelated RBC as controls. The authors have performed this relevant control in the Acute Pneumonia mouse model. The results presented in Supp Fig 9 show that the biodistribution of GM and gelated RBC is not the same, probably due to several factors such as the shape and the changes thereof, but also the fact that GM might have a specific tropism to inflammatory lungs.

In the third round, the previous reviewer questioned the fact that the gelated RBC were not stuck in the lung but rather accumulated in the liver. In response to this concern, the authors have argued that the role of the liver is to scavenge “damaged” RBC through phagocytosis by liver monocytes (ie, Kupffer’s cells). I totally agree on this argument as gelated RBC can be considered as damaged RBC.

One should also consider two other arguments : 1) gelated RBC are logically less deformable than healthy RBC, therefore they will have less ease to circulate through the smallest capillaries of the lungs, and 2) during lung inflammation, there is vasodilation of capillaries that will favour the “escape” of gelated RBC to the liver.

Based on the above analysis regarding specifically the previous reviewer’s concerns about the missing controls in the first submitted manuscript, I consider that the author have brought the expected results, and that from this particular point of view, the manuscript should be accepted for publication in Nature Communications.

Response to Review

Reviewer #4 (Remarks to the Author):

Months ago, Gao et al. submitted a manuscript entitled “Intracellularly gelled macrophages for targeted therapy of inflammatory diseases” to Nature Communications. This manuscript underwent several rounds of evaluation, and I have been specifically asked by the editor to assess whether “I consider that the concerns of reviewer #2 have been addressed. This is particularly in terms of the control experiments asked for”.

In first round, the previous reviewer asked for missing controls: 1) the comparison of the behaviour of RBC with the one of gelled macrophages (GM); and 2) the controls of DS+anti-TNF (related to Fig 5). The authors have properly addressed point 2). Regarding point 1), they have compared the behaviour of GM with the one of non-gelled RBC.

Therefore, in the second round, the reviewer logically asked for gelled RBC as controls. The authors have performed this relevant control in the Acute Pneumonia mouse model. The results presented in Supp Fig 9 show that the biodistribution of GM and gelled RBC is not the same, probably due to several factors such as the shape and the changes thereof, but also the fact that GM might have a specific tropism to inflammatory lungs. In the third round, the previous reviewer questioned the fact that the gelled RBC were not stuck in the lung but rather accumulated in the liver. In response to this concern, the authors have argued that the role of the liver is to scavenge “damaged” RBC through phagocytosis by liver monocytes (ie, Kupffer’s cells). I totally agree on this argument as gelled RBC can be considered as damaged RBC.

One should also consider two other arguments : 1) gelled RBC are logically less deformable than healthy RBC, therefore they will have less ease to circulate through the smallest capillaries of the lungs, and 2) during lung inflammation, there is vasodilation of capillaries that will favour the “escape” of gelled RBC to the liver.

Based on the above analysis regarding specifically the previous reviewer’s concerns about the missing controls in the first submitted manuscript, I consider that the author have brought the expected results, and that from this particular point of view, the manuscript should be accepted for publication in Nature Communications.

Response: Thanks for your kind recognition of our efforts in improving the overall quality of the manuscript during the past rounds, and for the kind recommendation of this work for publication.